# Single-molecule imaging reveals multiple pathways for the recruitment of translesion polymerases after DNA damage

Elizabeth S. Thrall[1], James E. Kath[1,2], Seungwoo Chang[1] & Joseph J. Loparo[1]

Unrepaired DNA lesions are a potent block to replication, leading to replication fork collapse, double-strand DNA breaks, and cell death. Error-prone polymerases overcome this blockade by synthesizing past DNA lesions in a process called translesion synthesis (TLS), but how TLS polymerases gain access to the DNA template remains poorly understood. In this study, we use particle-tracking PALM to image live *Escherichia coli* cells containing a functional fusion of the endogenous copy of Pol IV to the photoactivatable fluorescent protein PAm-Cherry. We find that Pol IV is strongly enriched near sites of replication only upon DNA damage. Surprisingly, we find that the mechanism of Pol IV recruitment is dependent on the type of DNA lesion, and that interactions with proteins other than the processivity factor β play a role under certain conditions. Collectively, these results suggest that multiple interactions, influenced by lesion identity, recruit Pol IV to sites of DNA damage.

[1] Department of Biological Chemistry and Molecular Pharmacology, Harvard Medical School, 250 Longwood Avenue, Boston, MA 02115, USA. [2] Present address: Discovery Chemistry and Technology, AbbVie Inc., 1 North Waukegan Road, North Chicago, IL 60064, USA. Correspondence and requests for materials should be addressed to J.J.L. (email: joseph_loparo@hms.harvard.edu)

Replicative DNA polymerases are extremely efficient and high-fidelity enzymes. They can be blocked, however, by the presence of unrepaired DNA damage on the template strand. This blockage can lead to replication fork collapse, double-strand DNA breaks, and ultimately cell death. Translesion synthesis (TLS) is one pathway for managing unrepaired DNA damage encountered during replication[1, 2]. In this process, a specialized TLS polymerase exchanges with a stalled replicative polymerase, extends the nascent DNA strand past the lesion, and then returns the template to the replicative polymerase for the continuation of normal synthesis. TLS polymerases, many of which are members of the Y-family of DNA polymerases, are typically error-prone and slow enzymes; their access to the template must therefore be tightly regulated[3].

Across all domains of life, misregulation of TLS can have severe consequences. In humans, mutations in the TLS polymerase Pol η give rise to the disorder xeroderma pigmentosum, in which sunlight sensitivity and skin cancer susceptibility are enhanced[3]. Increased expression of TLS polymerases, conversely, is common in a range of different cancer types and is thought to contribute to the mutator phenotype[4]; TLS polymerases may therefore represent a therapeutic target in cancer[5, 6]. In the model bacterium *Escherichia coli*, overproduction of TLS polymerases slows DNA replication and leads to cell death[7] and may contribute to the lethality of certain antibiotics[8].

*E. coli* has five DNA polymerases, three of which are TLS polymerases. Pol IV, a Y-family polymerase and homolog of human Pol κ, is the most abundant TLS polymerase in the cell. There are ~200 copies of Pol IV in normally growing *E. coli* cells, and Pol IV expression is induced further under conditions of stress. In particular, the SOS DNA damage response upregulates Pol IV levels 10-fold[9]. Pol IV is thought to be specialized for small minor groove lesions. In particular, it bypasses $N^2$ adducts of guanine ($N^2$-dG) of the type formed by the DNA damaging agent nitrofurazone (NFZ) with high efficiency and accuracy[10]; it also plays a role in bypassing alkyl adducts such as the $N^3$-methyladenine ($N^3$-mdA) lesions generated by methyl methanesulfonate (MMS)[11]. In contrast, the highly mutagenic TLS polymerase Pol V only accumulates late in the SOS response[3] and is able to bypass more severe forms of DNA damage, including major groove lesions[1].

How TLS polymerases gain access to the DNA template remains poorly understand. One model suggests that Pol IV remodels the replisome at a lesion site, exchanging dynamically with a stalled replicative polymerase to carry out TLS[12]. Alternatively, the replisome may skip over leading strand lesions, repriming downstream to continue replication. In this second model, TLS occurs behind the replication fork in a gap-filling reaction[13, 14]. In either case, however, it is clear that interactions with the sliding clamp processivity factor are essential for TLS in prokaryotes and eukaryotes[9], including TLS by Pol IV[15, 16]. All five *E. coli* polymerases interact with the β-clamp through conserved clamp-binding motifs (CBMs). Since β is dimeric and therefore has two polymerase-binding sites, it has been proposed to act as a molecular "toolbelt" that binds a replicative polymerase and a TLS polymerase simultaneously in order to facilitate polymerase exchange[15, 17].

We recently reconstituted TLS in vitro and observed polymerase exchange at the single-molecule level[18]. These experiments demonstrated that β can simultaneously bind Pol IV and Pol III, the *E. coli* replicative polymerase, and that polymerase exchange can occur through conformational dynamics of polymerases on the clamp. In cells, however, there are many other competitors for β-clamp binding. In addition to Pol III and Pol IV, at least 10 other proteins have been shown to bind β, including factors involved in Okazaki fragment maturation,

mismatch repair, and cell cycle regulation[9, 19]. Given the limited number of β-binding sites and the abundance of possible binding partners, Pol IV may not have access to the clamp when the replisome encounters a lesion. It is also not known whether previously identified interactions with other replication-associated and repair-associated proteins[20–22] play a role in recruiting Pol IV.

To better understand how access of Pol IV to the replisome is regulated in vivo, we have created a strain bearing a new functional fusion of the genomic copy of Pol IV to a photoactivatable fluorescent protein, PAmCherry. By imaging this strain using particle-tracking photoactivation localization microscopy (PALM), we have characterized the localization and dynamics of Pol IV in cells under normal growth conditions and after treatment with DNA damaging agents. We find that Pol IV is localized broadly throughout the cell during normal growth, with only modest β-dependent enrichment at replication forks. In the presence of DNA damage, however, Pol IV is strongly enriched at particular cellular positions. Surprisingly, the nature of the DNA damage has a profound impact on both the cellular localization of Pol IV and the recruitment mechanism. For MMS-treated cells, there is a high degree of Pol IV enrichment, which requires the well-characterized β-clamp interaction. In contrast, this interaction is not required for the majority of Pol IV localizations in NFZ-treated cells, indicating that Pol IV recruitment involves non-clamp interactions under some circumstances. Taken together, our results suggest that Pol IV recruitment is strongly influenced by the type of DNA lesion and that it involves multiple damage-induced interactions.

## Results

**Creation of a functional Pol IV-PAmCherry fusion.** To study the behavior of individual Pol IV molecules in live *E. coli* cells, we used the λ Red recombineering system[23] to replace the endogenous copy of Pol IV, encoded by the *dinB* gene, with a C-terminal fusion to the photoactivatable fluorescent protein PAmCherry (Fig. 1a). Because the C terminus of Pol IV is implicated in clamp-binding interactions[19], we introduced a 20-amino acid flexible Gly-Ser linker between Pol IV and PAmCherry. Cells bearing the Pol IV-PAmCherry fusion in the MG1655 wild-type (WT) strain background grew normally (Supplementary Note 3) and displayed close to WT levels of resistance to the DNA damaging agent nitrofurazone (NFZ) (Supplementary Fig. 1a). Consistent with previous reports[10, 16], cells lacking Pol IV (ΔdinB) were severely sensitized to NFZ. As expected, a Pol IV-PAmCherry fusion strain containing a Pol IV mutant that could not bind the clamp was as sensitized as the ΔdinB strain (Supplementary Fig. 1a), confirming that the PamCherry fusion does not prevent clamp binding.

Pol IV levels are upregulated ten-fold by the SOS response, which is induced by certain DNA damaging agents[9]. To increase the number of Pol IV molecules in the cell under normal growth conditions and to avoid possible complications caused by changes in the Pol IV level upon DNA damage, we chose to introduce the Pol IV-PAmCherry fusion into a strain that is constitutively activated for the SOS response. This strain bears the *lexA51* allele encoding a frameshift mutation of the LexA SOS repressor and the *sulA211* allele, which alleviates the block on cell division by SulA[24, 25]. Expression of Pol IV and other LexA-repressed gene products, including RecA and Pol V, in this ΔlexA strain are upregulated even in undamaged cells. As in the MG1655 background, the ΔlexA Pol IV-PAmCherry fusion strain had similar NFZ resistance to the ΔlexA strain with untagged Pol IV at 8 μM NFZ concentration (Fig. 1b) and slightly higher sensitivity at 10.5 μM NFZ concentration (Supplementary Fig. 1a).

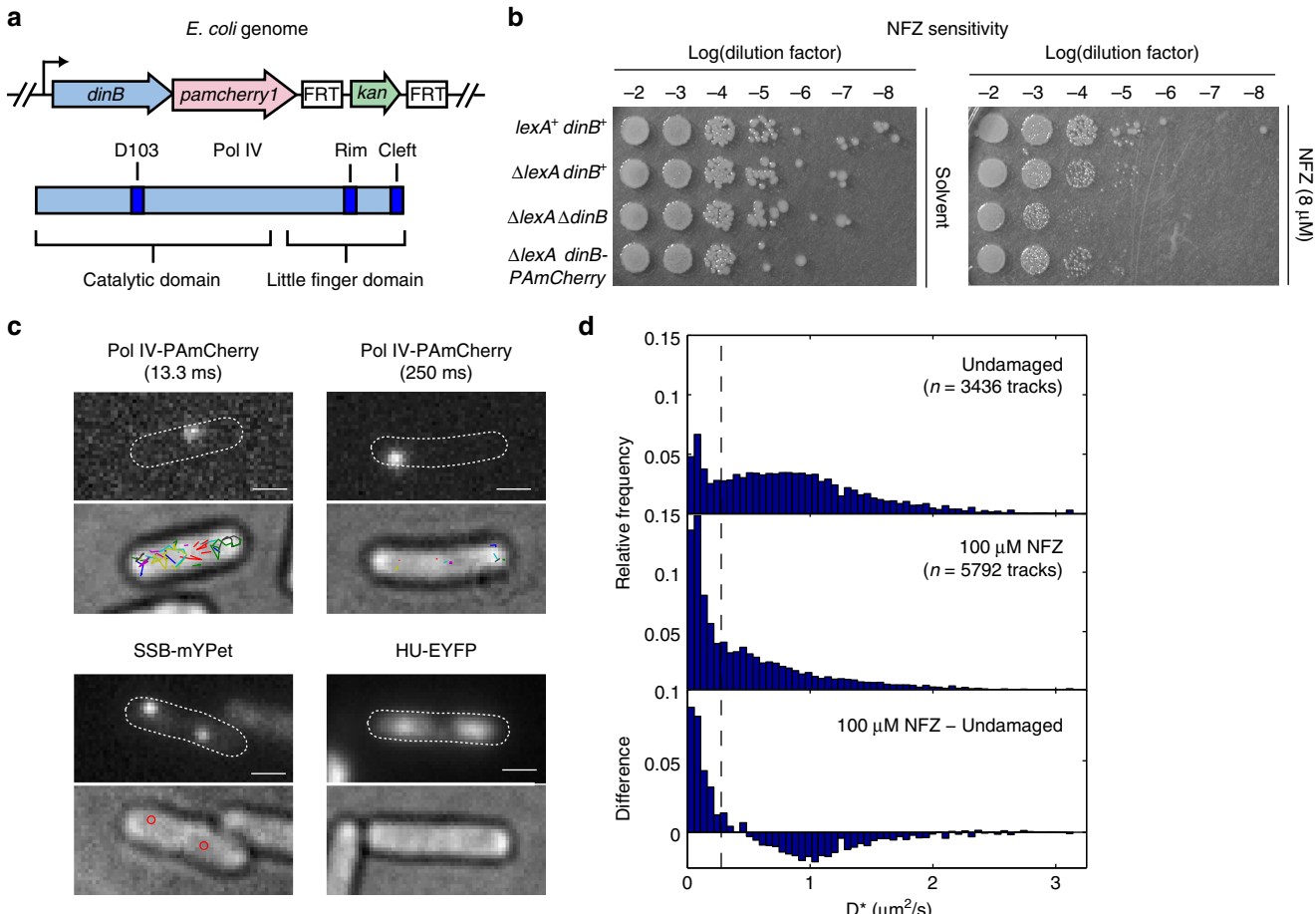

**Fig. 1** Particle-tracking PALM imaging of a functional Pol IV-PAmCherry fusion. **a** Schematic of the fusion of the photoactivatable fluorescent protein gene *pamcherry1* to the C terminus of the *dinB* gene encoding Pol IV (top) and schematic of the relevant Pol IV domains and residues (bottom). **b** Serial 10-fold dilutions of *E. coli* strains grown on LB agar plates without (left) and with (right) nitrofurazone (NFZ) added at 8 μM concentration. **c** Top panels: representative fluorescence micrographs of single activated Pol IV-PAmCherry molecules recorded with 13.3 ms (top left) and 250 ms (top right) integration times, SSB-mYPet foci (bottom left), and a nucleoid labeled with HU-EYFP (bottom right), with overlays of the cell outlines. Bottom panels: the corresponding brightfield micrographs with an overlay of all detected Pol IV-PAmCherry tracks (top left and top right) or all detected SSB-mYPet foci (bottom left) in the cell (Scale bars: 1 μm). **d** Distributions of the apparent diffusion coefficient ($D^*$) for Pol IV-PAmCherry in undamaged (top) and 100 μM NFZ-treated cells (middle), and the difference (bottom). The dashed lines indicate the threshold $D^*$ value for bound molecules ($D^* < 0.275 \, \mu m^2/s$)

**Static and mobile populations of Pol IV in undamaged cells**. To characterize the diffusive properties of Pol IV-PAmCherry in normally growing cells, we imaged cells using particle-tracking PALM microscopy with a short integration time of 13.3 ms, which allows for detection of rapidly diffusing proteins of similar size to Pol IV[26, 27]. We recorded movies using continuous low-intensity 405 nm and high-intensity 561 nm laser excitation. The low 405 nm photoactivation laser excitation stochastically converts individual PAmCherry molecules from a dark state to a bright state; activated molecules are then excited by the 561 nm illumination. Activated Pol IV-PAmCherry molecules appeared as bright spots (Fig. 1c). We determined the position of Pol IV-PAmCherry molecules in each frame by fitting a 2D Gaussian approximation of the point spread function (PSF) to the fluorescent spots and then linked detected spots to form trajectories.

We calculated an apparent two-dimensional diffusion coefficient $D^*$ for each Pol IV-PAmCherry trajectory as: $D^* = \frac{MSD}{4\Delta t}$. This value does not represent the true diffusion coefficient because it reflects localization error, motion blurring, and cell confinement effects[28]. Nonetheless, we could resolve two populations of Pol IV in normally growing cells (Fig. 1d). One population, with $D^* \approx 0.8 \, \mu m^2/s$, represents mobile Pol IV molecules. This value is in reasonable agreement with the

apparent diffusion coefficients observed for PAmCherry fusions to other *E. coli* proteins of similar size[26]. The second population, with $D^* \approx 0.1 \, \mu m^2/s$, represents Pol IV molecules that are statically bound, likely to DNA. By imaging Pol IV-PAmCherry in cells fixed with formaldehyde, we could set a threshold $D^*$ value of 0.275 $\mu m^2/s$ to identify static molecules (Supplementary Fig. 1b). We found that ~21% of Pol IV molecules fell below this $D^*$ value under normal growth conditions.

**Localization of static Pol IV molecules in undamaged cells**. Pol IV-PAmCherry molecules carrying out TLS must be statically bound to DNA. Therefore, we switched to a mode of imaging that allowed us to resolve static molecules selectively and over longer timescales. We increased the integration time to 250 ms and made a concurrent reduction in the excitation power. This long exposure time blurred out the fluorescence signal from mobile molecules, while statically bound molecules still appeared as sharp foci (Fig. 1c and Supplementary Fig. 2a). We identified static molecules based on their PSF width by comparison to the distribution of PSF widths in fixed cells (Supplementary Fig. 2b).

To characterize the cellular localization of static Pol IV molecules, we determined the cell outline from the brightfield

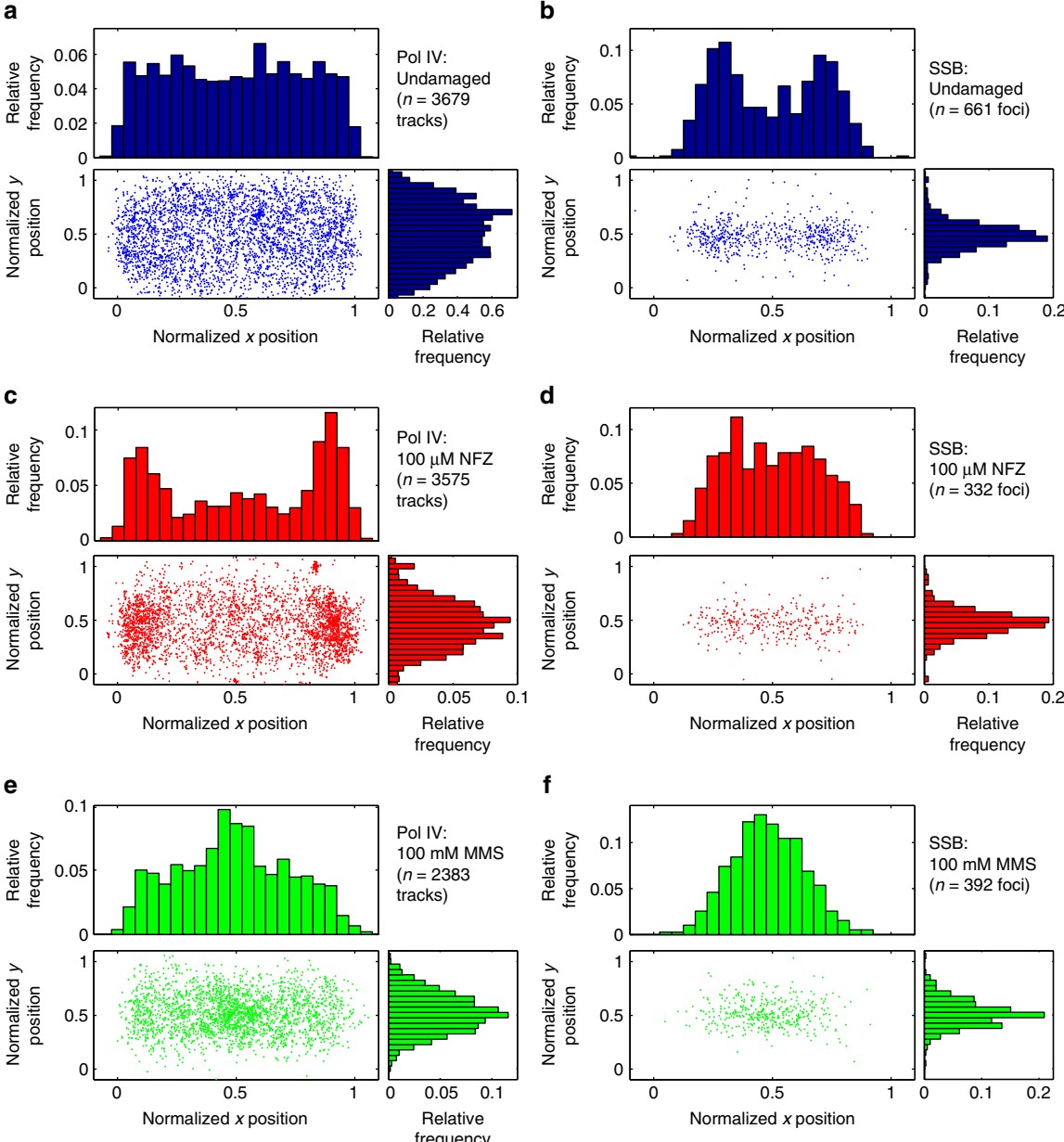

**Fig. 2** Cellular localization of Pol IV-PAmCherry and SSB-mYPet in undamaged, NFZ-treated, and MMS-treated cells. Scatter plots of normalized mean coordinates of Pol IV-PAmCherry tracks and the normalized coordinates of SSB-mYPet foci in undamaged (**a**, **b**), 100 μM NFZ-treated (**c**, **d**), and 100 mM MMS-treated (**e**, **f**) cells. Also shown are the projections of the scatter plots along the long (x) and short (y) cellular axes

micrograph and normalized all positions along the long (x) and short (y) cell axes to range from 0 to 1 (Supplementary Fig. 2c). For each Pol IV trajectory, we calculated the mean position along these two axes. A scatter plot of Pol IV positions across many cells shows that Pol IV was localized throughout the cell under normal growth conditions (Fig. 2a). Projections of this distribution along the long and short cell axes revealed no Pol IV enrichment at any cellular position. To determine the position of replication forks in the same cells, we introduced a second copy of single-stranded DNA-binding protein (SSB) fused to mYPet (Fig. 1c), an established replisome marker[29] that enables orthogonal two-color imaging. In contrast to Pol IV, SSB-mYPet foci were strongly localized at the ¼ and ¾ long-axis positions (Fig. 2b), consistent with previous reports for the position of replication foci in slowly growing *E. coli* cells[29, 30].

To assess Pol IV-SSB colocalization on a single-cell level, we calculated the mean distance between each static Pol IV track and the nearest SSB focus, yielding a broad multimodal distribution with a peak near 370 nm (Fig. 3a). To quantify Pol IV enrichment near SSB foci, we normalized this distribution by a simulated random distribution (Methods section) to generate the radial distribution function, $g(r)$[31, 32]. This function expresses the likelihood of Pol IV localization at a distance $r$ from SSB relative to random cellular localization, with values of $g(r) > 1$ indicating enrichment. Using this approach, we reproduced a previous report for *E. coli* ParC-PAmCherry and MukB-mYPet[31], which are modestly colocalized (Supplementary Fig. 3b). Demonstrating that we can detect colocalization of proteins bound at small copy numbers, we observed enrichment of a PAmCherry fusion to the ε exonuclease subunit of the replicative polymerase Pol III near SSB foci (Supplementary Fig. 3d). Finally, we showed that strong

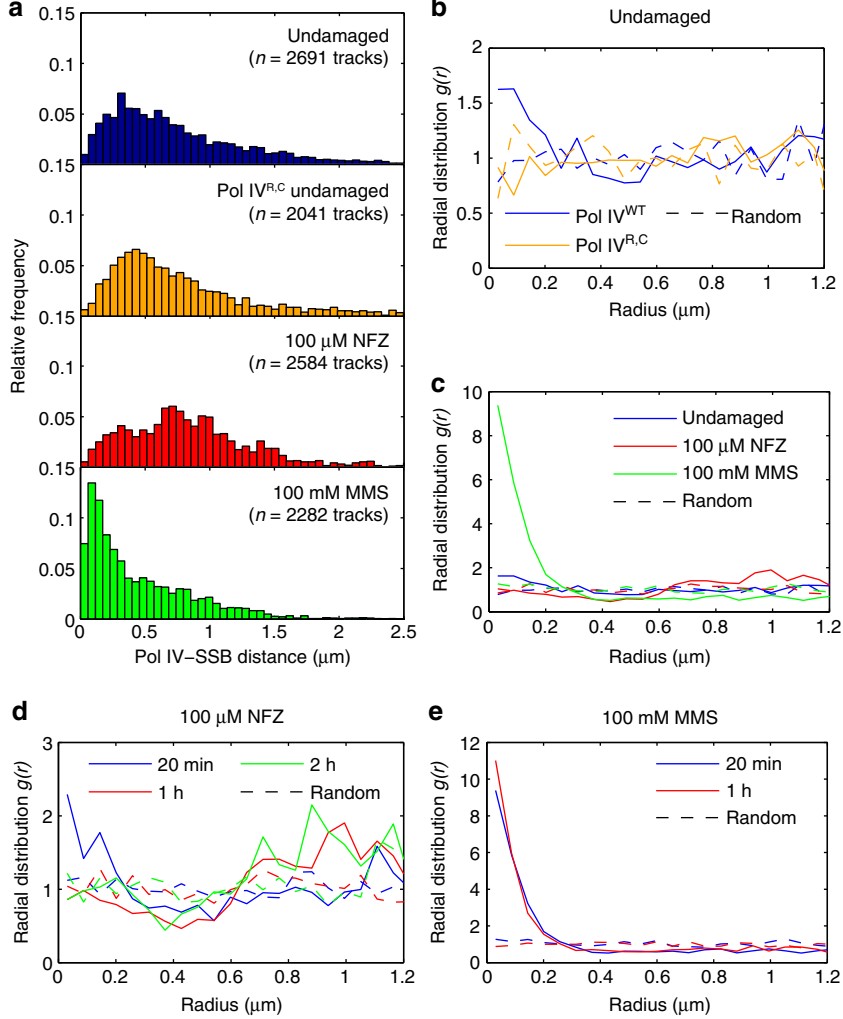

**Fig. 3** Single-cell colocalization of Pol IV-PAmCherry and SSB-mYPet in undamaged, NFZ-treated, and MMS-treated cells. **a** Distributions of the mean distance between each static Pol IV track and the nearest SSB focus for Pol IV[WT] (top) and Pol IV[R,C] (top center) in undamaged cells and for Pol IV[WT] in cells treated with 100 μM NFZ (bottom center) or 100 mM MMS (bottom). **b** Radial distribution functions $g(r)$ for the undamaged distance distributions in **a**. **c** Radial distribution functions $g(r)$ for the damaged Pol IV[WT] distance distributions in **a**. **d** Radial distribution functions $g(r)$ for Pol IV[WT] in cells treated with 100 μM NFZ for 20 min, 1 h, or 2 h. **e** Radial distribution functions $g(r)$ for Pol IV[WT] in cells treated with 100 mM MMS for 20 min or 1 h. Also shown in panels **b**–**e** are random $g(r)$ functions for each data set. Several Pol IV[WT] $g(r)$ traces are replotted to enable comparison

nucleoid localization alone does not produce strong apparent colocalization with replication forks by determining $g(r)$ for a PAmCherry fusion to the DNA-binding protein HU[33]; in this case, $g(r)$ peaks at an intermediate separation distance and approaches a value of one as $r$ approaches zero (Supplementary Fig. 3d). Consistent with the average cellular localization profiles, this $g(r)$ analysis revealed that Pol IV is only weakly colocalized with SSB foci, and therefore replication forks, in undamaged cells, with $g(r) \approx 1.5$ at short distances (Fig. 3b).

Next we asked whether this weak colocalization was dependent on interactions with the β-clamp. Pol IV contains two domains, an N-terminal catalytic domain (residues 1–230) and a C-terminal little finger domain (residues 243–351), connected by a flexible linker (Fig. 1a). The little finger domain of Pol IV makes two separate contacts to β, one to a cleft on the face of one protomer and a second to the clamp rim near the dimer interface.[19, 34] Mutation of these two sets of residues (see Supplementary Note 2 for strain JEK717) yields a mutant (Pol IV[R,C]) that is deficient for clamp binding. In contrast to the weak enrichment of WT Pol IV (Pol IV[WT]), there was no enrichment of Pol IV[R,C] near SSB foci in undamaged cells (Fig. 3b). Taken

together, these results indicate that Pol IV is only modestly enriched near replication forks in the absence of exogenous DNA damage, and that this enrichment is mediated by interactions with the β-clamp.

**NFZ treatment alters the localization of static Pol IV.** We next characterized the behavior of Pol IV in cells treated with the DNA-damaging agent NFZ in liquid cultures. First, we assessed how 1 h exposure to 40 μM or 100 μM NFZ affected cell viability in liquid culture. There was a modest increase in the number of viable cells after treatment with 40 μM NFZ, although growth was slowed. In contrast, we found a moderate decrease in cell viability after 100 μM NFZ treatment; there was, however, no difference between strains with or without the PAmCherry fusion (Supplementary Note 4). As expected, western blot analysis of a C-terminally FLAG-tagged version of the Pol IV-PAmCherry fusion confirmed that NFZ treatment did not further induce Pol IV expression in the Δ*lexA* strain (Supplementary Fig. 4a and Supplementary Note 5).

NFZ-treated cells were prepared for microscopy and imaged in the same way as undamaged cells. Using short exposures first to

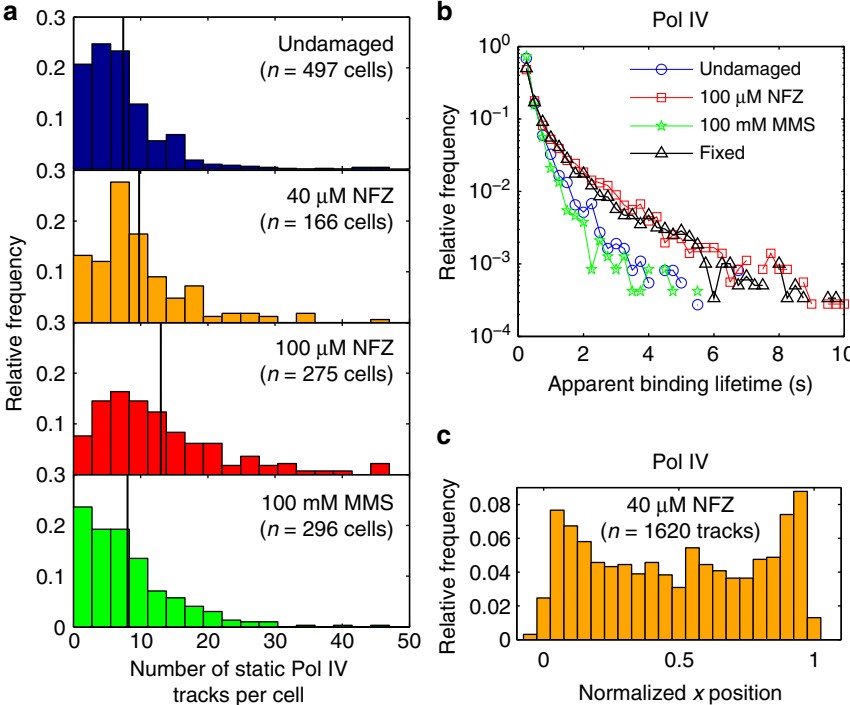

**Fig. 4** Effect of NFZ and MMS treatment on the number, lifetime, and cellular localization of static Pol IV-PAmCherry tracks. **a** Distributions of the number of static Pol IV-PAmCherry tracks per cell in undamaged cells and cells treated with 40 μM NFZ, 100 μM NFZ, or 100 mM MMS. The mean of each distribution is indicated by a solid line. **b** Distribution of the apparent Pol IV-PAmCherry binding lifetime in undamaged cells (blue circles) and cells treated with 100 μM NFZ (red squares) or 100 mM MMS (green stars). Also shown is the apparent Pol IV-PAmCherry binding lifetime in fixed cells (black triangles) as a photobleaching control. **c** Long-axis cellular localization of Pol IV-PAmCherry in cells treated with 40 μM NFZ

look at Pol IV diffusion, we observed an increase in the static fraction of trajectories from ~21 to 47% after treatment with 100 μM NFZ (Fig. 1d). Subtraction of the undamaged $D^*$ distribution from the NFZ-treated $D^*$ distribution revealed a decrease in the relative share of the mobile Pol IV population and a corresponding increase in the static population (Fig. 1d). In a strain lacking the PAmCherry fusion, we saw only a small number of false positive trajectories, primarily with low $D^*$ values, for both undamaged and NFZ-treated cells (Supplementary Fig. 1c).

Under imaging conditions employing long integration times, we observed a dose-dependent increase in the number of static Pol IV molecules per cell upon NFZ treatment. The mean number of static tracks increased from $7.4 \pm 0.3$ (mean ± s.e.m.) in undamaged cells to $9.8 \pm 0.6$ in 40 μM NFZ-treated cells to $13.0 \pm 0.6$ in 100 μM NFZ-treated cells under matched imaging conditions ($p < 10^{-3}$ for undamaged vs. 40 μM NFZ and $p < 10^{-4}$ for 40 μM vs. 100 μM NFZ; Fig. 4a). From these numbers, we estimate that ~13 and 22% of Pol IV molecules are static in undamaged and 100 μM NFZ-treated cells, respectively (Supplementary Note 6). In control experiments, treatment with the solvent used to prepare the NFZ solution did not cause an increase in the number of static Pol IV tracks, and we saw very few localizations in a strain containing only SSB-mYPet without the Pol IV-PAmCherry fusion for both undamaged and 100 μM NFZ-treated cells (Supplementary Fig. 5a,b).

We next asked whether DNA damage affected the Pol IV binding time, which we determined by measuring the length of static Pol IV trajectories. Trajectories end when a Pol IV-PAmCherry molecule blinks, photobleaches irreversibly, or unbinds. Thus the apparent binding lifetime that we measure reflects both PAmCherry photophysics and the Pol IV binding lifetime. Even with these limitations, comparison of the binding

lifetime distributions (Fig. 4b) revealed that Pol IV remained bound significantly longer in 100 μM NFZ-treated cells than in undamaged cells ($p \ll 10^{-5}$). The binding lifetimes in NFZ-treated and fixed cells, however, were similar (<10% difference in means, $0.01 < p < 0.05$), indicating that our measurement of the Pol IV binding lifetime in NFZ-treated cells was limited by PAmCherry photobleaching.

In addition to these differences in Pol IV binding, there was a dramatic change in the cellular localization of Pol IV in NFZ-treated cells, with strong enrichment at positions ~10–15% and 85–90% along the long cell axis (Fig. 2c); like the dose-dependent increase in the number of static Pol IV tracks, the localization pattern became more pronounced at the higher NFZ concentration (compare Fig. 4c and Fig. 2c). The localization of Pol IV along the short cell axis remained peaked at midcell, although its distribution was somewhat narrower. SSB localization, by comparison, was peaked at the 1/3 and 2/3 long-axis positions, with more density at midcell than in undamaged cells (Fig. 2d), likely reflecting a 16% decrease in mean cell length upon NFZ treatment (Supplementary Fig. 6a,b) and the tendency of SSB to be localized closer to midcell in smaller cells (Supplementary Fig. 7). Pol IV was shifted toward the cell pole relative to SSB in NFZ-treated cells. Consistent with the average cellular localization profiles, single-cell colocalization analysis confirmed that Pol IV was not strongly colocalized with SSB foci in NFZ-treated cells. Instead, there was a 13% increase in the average Pol IV-SSB distance ($p \ll 10^{-5}$) in comparison to undamaged cells (Fig. 3a), and the radial distribution function $g(r)$ revealed no enrichment at short distances (Fig. 3c).

The characteristic Pol IV localization pattern was established by 20 min after NFZ addition and persisted for at least 2 h (Supplementary Fig. 8a,b). The somewhat weaker response at the 20 min time point may reflect the fact that NFZ is activated after

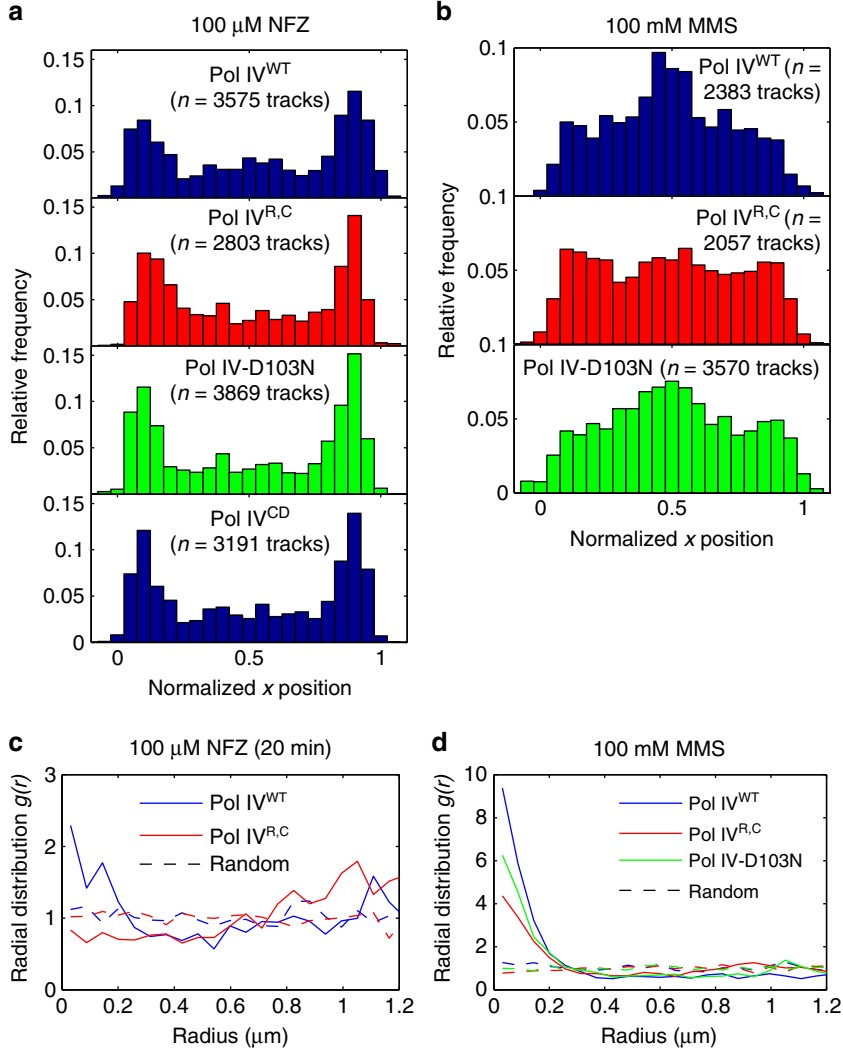

**Fig. 5** Long-axis cellular localization profiles and single-cell colocalization analysis of Pol IV mutants. **a** Long-axis cellular localization of Pol IV[WT] and the Pol IV[R,C], Pol IV-D103N, and Pol IV[CD] mutants in cells treated with 100 µM NFZ. **b** Long-axis cellular localization of Pol IV[WT] and the Pol IV[R,C] and Pol IV-D103N mutants in cells treated with 100 mM MMS. **c** Radial distribution functions $g(r)$ for Pol IV[WT] and the Pol IV[R,C] mutant in cells treated with 100 µM NFZ for 20 min. **d** Radial distribution functions $g(r)$ for Pol IV-SSB distance for Pol IV[WT] and the Pol IV[R,C] and Pol IV-D103N mutants in cells treated with 100 mM MMS. Also shown in panels **c** and **d** are random $g(r)$ functions for each data set. The Pol IV[WT] data in **a–d** are replotted from Fig. 2c, e and Fig. 3c, d to enable comparison

processing by cellular nitroreductases[35, 36], and thus it may take some time for DNA damage to accumulate. Interestingly, there was weak Pol IV-SSB colocalization at the 20 min time point, with a very modest increase in $g(r)$ relative to undamaged cells, but enrichment was lost by 1 h (Fig. 3d and Supplementary Fig. 9a). As for undamaged cells, the weak Pol IV-SSB colocalization at the 20 min time point was dependent on clamp-binding, with no enrichment observed for the Pol IV[R,C] mutant (Fig. 5c and Supplementary Fig. 9c).

The distinct Pol IV localization pattern observed after NFZ treatment was found across the whole population of cells, not just in a small fraction of potentially non-viable cells. Comparison of the cellular localization of Pol IV in two populations sorted by the number of static tracks per cell shows that the Pol IV localization pattern was qualitatively the same for cells in the top 25% and the bottom 75% of the population (Supplementary Fig. 10). We observed a similar Pol IV localization pattern in the isogenic *lexA*[+] strain, indicating that Pol IV recruitment is not unique to the fully SOS-induced state (Supplementary Fig. 11a). To confirm that the Pol IV localization was not due to fluorescent protein

oligomerization[37], we tested a Pol IV fusion to a different photoactivatable fluorescent protein, mMaple3, which was found to have undetectable levels of dimerization in a recent study[33]. Although this fusion appeared to be less functional than the corresponding PAmCherry fusion (Supplementary Fig. 1a), the cellular localization was very similar in both undamaged and NFZ-treated cells (Supplementary Fig. 11a). Furthermore, we saw substantially fewer static localizations in a strain where the *dinB* gene was replaced with PAmCherry (Supplementary Fig. 5a,b), indicating that the effects we observe are Pol IV dependent.

Next, we investigated whether other polymerases were recruited to a similar cellular position upon NFZ treatment by imaging strains bearing PAmCherry fusions to either Pol I or the ε exonuclease subunit of the replicative polymerase Pol III under the same conditions. We found that Pol I molecules bound randomly throughout the nucleoid in both undamaged and NFZ-treated cells (Supplementary Figure 11b), with no strong enrichment at any particular cellular position. These results are consistent with a previous finding that the localization of Pol I did not change significantly upon treatment with MMS[26].

Interestingly, we found that ε-PAmCherry was recruited to a similar cellular position as Pol IV in NFZ-treated cells, although less strongly (Supplementary Fig. 11b), raising the possibility that these areas contain DNA replication intermediates.

To determine how the replisome was affected by NFZ treatment, we analyzed the response of several replisome markers. In addition to SSB-mYPet, both ε-mYPet and a previously characterized YPet-β fusion[30, 38] mark replication forks as expected (Supplementary Fig. 12a), with most cells containing 1 or 2 foci. There were modest decreases in the number of replisome foci per cell after NFZ treatment (Supplementary Fig. 12b and Supplementary Table 4), but the results suggest that replication was not completely disrupted. Analysis of the mean intensity of replisome foci (Supplementary Table 5 and Methods section) reveal that the intensity of SSB and ε foci more than doubled, whereas the intensity of β foci increased by a smaller amount; these effects held when comparing cells with the same number of replisome foci (Supplementary Fig. 13) to minimize possible complications from differences in the populations of cells.

We also characterized the nucleoid morphology by imaging a plasmid-borne EYFP fusion to the DNA-binding protein HU (Fig. 1c)[39]. Inspection of the average HU intensity profiles (Supplementary Fig. 14a and Methods section) revealed that there was very little change in the nucleoid extent or in the HU fluorescence intensity upon treatment with 100 μM NFZ (Supplementary Fig. 14b), with the minor differences caused by the smaller average cell size in NFZ-treated cells (Supplementary Fig. 14c). Therefore, the change in Pol IV localization is not due to dramatic reorganization of the nucleoid. Comparison of the nucleoid profile to the cellular localization of Pol IV-PAmCherry in NFZ-treated cells revealed that Pol IV was localized toward the periphery of the nucleoid (Supplementary Fig. 14d).

**MMS treatment alters the localization of static Pol IV.** Next we attempted to determine whether the observed Pol IV recruitment was a general phenomenon in damaged cells or specific to NFZ treatment. We exposed cells to MMS, an alkylating agent that generates DNA lesions that can be bypassed by Pol IV[40]. We incubated cells on agarose pads containing 100 mM MMS for 20 min, a condition that was shown to activate base excision repair without being lethal[26]. In MMS-treated cells, Pol IV localized to the midcell position along the long axis (Fig. 2e), in contrast to its localization upon NFZ treatment. The localization pattern of SSB foci was sharply peaked at midcell (Fig. 2f), which may in part reflect the 25% reduction in cell length for MMS-treated cells (Supplementary Figure 6a,b and Supplementary Figure 7), and more strongly overlapped with that of Pol IV. Single-cell colocalization analysis revealed a 42% decrease in the mean Pol IV-SSB distance after MMS treatment (Fig. 3a; $p \ll 10^{-5}$), with a peak in the Pol IV-SSB distance distribution at ~115 nm. As the SSB foci are larger than diffraction-limited PSFs, this 115 nm offset is well within the average focus width, indicating colocalization of Pol IV and SSB. The dip at Pol IV-SSB separation distances close to zero reflects a number of factors, including the localization error for PAmCherry and mYPet (Methods section), and the measured separation cannot therefore be interpreted as the actual distance of Pol IV from the center of replication forks. Strikingly, radial distribution analysis revealed a strong increase in Pol IV enrichment near SSB foci, with a 6-fold higher maximum $g(r)$ value relative to undamaged cells (Fig. 3c).

The Pol IV localization pattern was stable for at least 1 h after exposure to MMS (Supplementary Fig. 8c), with an additional modest increase in $g(r)$ (Fig. 3e and Supplementary Fig. 9b). As a control experiment, we confirmed that MMS treatment did not

cause significant false-positive localizations in a strain containing only SSB-mYPet without the Pol IV-PAmCherry fusion (Supplementary Fig. 8d), nor was the observed localization due to the smaller size of MMS-treated cells (Supplementary Fig. 7). Despite the change in Pol IV localization upon MMS treatment, there was only a small and statistically insignificant increase in the number of static Pol IV tracks per cell (mean ± s.e.m. of $8.1 \pm 0.5$ in MMS-treated cells, not significant (NS) vs. undamaged cells) (Fig. 4a). In contrast to the increased Pol IV binding lifetime in NFZ-treated cells, there was a small but statistically significant decrease in the lifetime in MMS-treated cells relative to undamaged cells (Fig. 4b; 13% decrease in mean lifetime, $p < 10^{-4}$). As for NFZ treatment, we confirmed that MMS treatment led to only a minor reduction in the mean number of SSB-mYPet, ε-mYPet, and YPet-β foci per cell (Supplementary Fig. 12b), suggesting that replication was not severely impaired. The mean intensity of SSB foci did not change after MMS treatment, whereas the intensity of ε and β foci increased modestly (Supplementary Table 5 and Supplementary Fig. 13). The nucleoid profile in MMS-treated cells was narrower and peaked at midcell (Supplementary Fig. 14b), partially due to cell size effects (Supplementary Fig. 14c), but was not indicative of dramatic reorganization of the chromosome.

**Pol IV recruitment differs in NFZ- and MMS-treated cells.** Next we investigated what interactions were responsible for the localization of Pol IV in the presence of DNA damage. Because β-clamp contacts are known to be essential for TLS by Pol IV[15, 16], we first looked at the effect of eliminating the Pol IV–clamp interaction. Consistent with the importance of this interaction, the Pol IV recruitment in MMS-treated cells required clamp binding. Enrichment at midcell was clearly abolished for the Pol IV$^{R,C}$ mutant (Fig. 5b). Single-cell colocalization analysis revealed a 33% increase in the mean Pol IV-SSB distance for Pol IV$^{R,C}$ in comparison to Pol IV$^{WT}$ and attenuation of the peak at approximately 115 nm in the Pol IV$^{WT}$ distance distribution (Supplementary Fig. 9d; $p \ll 10^{-5}$), with a 2-fold reduction in the maximum $g(r)$ value (Fig. 5d). As expected, there were also fewer static Pol IV tracks in MMS-treated cells (Supplementary Figure 4d; mean ± s.e.m. of $5.1 \pm 0.2$ for Pol IV$^{R,C}$ vs. $8.1 \pm 0.5$ for Pol IV$^{WT}$, $p < 10^{-5}$). Next, we investigated whether Pol IV catalysis was required for recruitment by imaging a strain bearing a catalytically inactive Pol IV mutant, D103N. Although there was no change in the number of static Pol IV tracks in MMS-treated cells (Supplementary Fig. 4d; mean ± s.e.m. of $7.0 \pm 0.3$ for Pol IV-D103N, NS vs. Pol IV$^{WT}$), and the characteristic midcell localization was still present, we observed a modest decrease in Pol IV-SSB colocalization (Fig. 5d and Supplementary Fig. 9d) and a modest increase in the Pol IV binding lifetime (Supplementary Fig. 15c; 13% increase in mean lifetime, $p < 10^{-3}$).

Surprisingly, we found no effect of the clamp-binding mutations on the number of static Pol IV tracks in cells treated with 100 μM NFZ for 1 h (Supplementary Table 6 and Supplementary Fig. 4b; mean of $13.4 \pm 0.8$ for Pol IV$^{R,C}$ vs. $13.0 \pm 0.6$ for Pol IV$^{WT}$, NS), although there were fewer localizations in undamaged cells. Likewise, the characteristic localization pattern in NFZ-treated cells was equivalent to that of the WT protein (Fig. 5a), indicating that the observed Pol IV recruitment was independent of β. To determine what domains and what other interactions might be responsible for Pol IV recruitment in NFZ-treated cells, we constructed strains with the truncated N-terminal catalytic domain (Pol IV$^{CD}$) or the C-terminal little finger domain (Pol IV$^{LF}$) fused to PAmCherry (Fig. 5, Supplementary Fig. 4b, and Supplementary Table 6). Western blot analysis (Supplementary Fig. 4a) of the

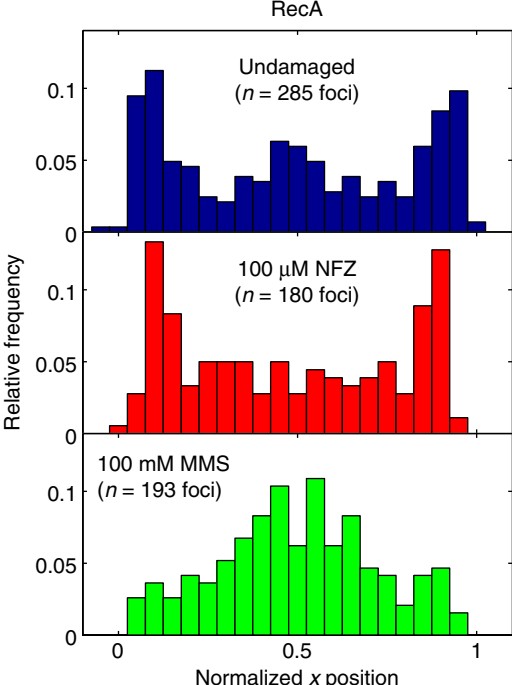

**Fig. 6** Effect of NFZ and MMS treatment on RecA cellular localization. Long-axis cellular localization of RecA-GFP foci in undamaged cells (top) and cells treated with 100 μM NFZ (middle) or 100 mM MMS (bottom)

corresponding FLAG-tagged fusions confirmed that the full-length Pol IV$^{CD}$ construct was expressed. The Pol IV$^{LF}$ fusion, however, appeared to be completely cleaved and failed to localize in cells (Supplementary Fig. 4a,c). The Pol IV$^{CD}$ mutant behaved similarly to Pol IV$^{WT}$ in NFZ-treated cells, although, as for the Pol IV$^{R,C}$ mutant, there were fewer localizations in undamaged cells. Given that the Pol IV catalytic domain was sufficient for localization, we tested whether Pol IV catalytic activity was also required by imaging the Pol IV-D103N mutant. The binding and localization of Pol IV-D103N was very similar to that of Pol IV$^{WT}$ in both undamaged and NFZ-treated cells, indicating that the Pol IV recruitment which we observe does not require catalytic activity. Consistent with the similar Pol IV localization patterns in NFZ-treated cells, the binding lifetimes were very similar for Pol IV$^{WT}$ and the mutant proteins, with the exception of a modest increase for Pol IV$^{CD}$ (Supplementary Fig. 15b; 19% increase in mean lifetime, $p < 10^{-5}$).

**Pol IV recruitment may involve RecA but not UmuD.** Although interactions with the β clamp are critical for TLS, Pol IV has also been shown to interact with a complex formed by RecA and the Pol V subunit UmuD[21], [22]. To determine whether interactions with UmuD play a role in the observed Pol IV localization upon DNA damage, we deleted the *umuDC* operon. In addition to eliminating any possible Pol IV-UmuD interaction, deletion of this operon also means that the TLS polymerase Pol V, comprised of the processed UmuD$'_2$ dimer and UmuC, will not be present. We found no change for the Δ*umuDC* strain in the number of static Pol IV tracks (Supplementary Fig. 16a,c and Supplementary Table 7) or the Pol IV cellular localization pattern (Supplementary Fig. 16b,d) in cells treated with NFZ and MMS. Therefore, Pol IV recruitment does not require interactions with UmuD, nor is it affected by the absence of Pol V. This finding is consistent with a previous study that looked at Pol IV-EYFP foci formed in response to different kinds of DNA damage and observed no effect upon deletion of *umuDC*.[41]

RecA plays a critical role in both DNA double-strand break repair and the cellular response to DNA damage, and RecA-deficient cells are highly sensitized to NFZ[42] and MMS treatment[43], [44]. For that reason, we chose to look for possible Pol IV−RecA interactions by determining the cellular localization of RecA rather than deleting it. We introduced a previously characterized RecA-GFP fusion[45] into the Δ*lexA* strain background and looked at its behavior in undamaged and damaged cells. In agreement with the initial report, RecA forms foci even in undamaged cells (Supplementary Fig. 17a), and these foci are primarily localized toward the cell poles (Fig. 6) in a similar position to that of Pol IV in NFZ-treated cells. Upon NFZ treatment, we found that the RecA localization pattern remained largely unchanged (Fig. 6), and there was a 36% decrease in the number of RecA foci per cell, from a mean ± s.e.m. of 1.38 ± 0.06 foci in undamaged cells to 0.92 ± 0.05 foci in NFZ-treated cells (Supplementary Fig. 17b; $p \ll 10^{-5}$). In MMS-treated cells, however, there was a marked change in the cellular localization of RecA. As previously observed for UV-irradiated cells[45], RecA foci were localized at the midcell position along the long cell axis (Fig. 6), in the same region of the cell as Pol IV and SSB. As for Pol IV, this change in localization pattern does not appear to be exclusively due to the decrease in average cell size (Supplementary Figure 7). Consistent with the results for NFZ treatment, MMS treatment did not cause a dramatic increase in the number of RecA foci. Instead there was a 50% decrease in the average number of RecA foci per cell after MMS treatment, from a mean ± s.e.m. of 1.38 ± 0.06 foci per cell in undamaged cells to 0.70 ± 0.04 in MMS-treated cells (Supplementary Fig. 17b; $p \ll 10^{-5}$).

## Discussion

In this study, we created a novel and functional PAmCherry fusion to the endogenous copy of Pol IV and used particle-tracking PALM to visualize the localization and dynamics of single Pol IV molecules in live *E. coli* cells. Consistent with studies of other DNA and RNA polymerases[26], [46], we observed two populations of Pol IV in cells, one diffusing and one statically bound. Selective imaging of static molecules revealed that Pol IV binds widely throughout the cell under normal growth conditions. A previous study, using a Pol IV-EYFP construct over-expressed from a plasmid, did not observe Pol IV binding in undamaged cells[41]; unlike PALM, however, this imaging technique is not sensitive to short binding events of single molecules. Importantly, we did not observe strong Pol IV localization in the vicinity of replication forks marked by SSB-mYPet. Instead, radial distribution function analysis revealed weak enrichment of Pol IV near SSB foci; this enrichment was dependent on interactions with the β-clamp. Given that there is a steady-state level of ~50 clamps near the fork during replication in *E. coli*[30], our results suggest that, in the absence of DNA damage, the average occupancy of Pol IV on these β-clamps is relatively low. This finding is consistent with the model that the access of Pol IV to the DNA template is tightly controlled under normal growth conditions in order to minimize mutagenesis[47].

A prior study reported the formation of multi-copy Pol IV foci in response to treatment with the DNA damaging agent 4-nitroquinoline 1-oxide (4-NQO) or double-strand break induction[41]. In this work, we found that DNA damage by NFZ or MMS led to strong Pol IV enrichment at particular cellular positions, with the nature of the enrichment dependent on the type of damage. In MMS-treated cells, we observed strong Pol IV recruitment to replication forks marked by SSB foci, whereas Pol IV binding was not enriched near SSB foci in NFZ-treated cells, except modestly at an early timepoint after NFZ exposure.

Corresponding to these differences in the pattern of Pol IV enrichment, we found differences in the Pol IV recruitment mechanism in NFZ-treated and MMS-treated cells. In MMS-treated cells, localization of a β-binding-deficient Pol IV mutant was attenuated, with a reduction in Pol IV-SSB colocalization and in the number of Pol IV-binding events, indicating that interactions with the clamp were necessary for the observed recruitment under those conditions. Surprisingly, we found that clamp-binding was not required for Pol IV recruitment in NFZ-treated cells, nor was Pol IV catalytic activity. Instead, the Pol IV N-terminal catalytic domain was sufficient for recruitment. We note that localization alone is not sufficient for TLS; the Pol IV[R,C], Pol IV-D103N, and Pol IV[CD] mutants that localize in NFZ-treated cells are nonetheless deficient for TLS[10, 16].

Why might there be differences in the response of Pol IV to DNA damage caused by NFZ and MMS? One key distinction between the two compounds is their differential induction of the SOS response, which is suggestive of different effects on replication. At drug concentrations to which ΔdinB cells are sensitized, MMS was found to induce the SOS response more strongly than NFZ[48]. Because SOS induction is triggered by persistent ssDNA tracts, this result implies that MMS lesions block replication more potently. Characterization of YPet-β foci before and after MMS treatment revealed a 61% increase in the average number of clamps at the fork; these additional clamps may be left behind at lesion sites after the replisome has stalled. It is unlikely that the relatively modest increase in the β copy number after MMS treatment is enough to explain the observed recruitment if mass action is the only determinant; instead, these results suggest that the access of Pol IV to the clamp is regulated in response to DNA damage, perhaps through conformational dynamics of Pol III[49, 50] or activation by RecA[51]. In contrast to the increased Pol IV binding lifetime in NFZ-treated cells, the slight decrease in lifetime upon MMS treatment shows either that many of the localizations that we observe represent transient clamp-binding without TLS or that Pol IV does not carry out extensive synthesis under these conditions. Consistent with this picture, we find only a modest increase in the binding lifetime of the Pol IV-D103N mutant. Previous work proposed that Pol IV-D103N becomes "locked" at a DNA lesion, unable to complete the catalytic cycle[16]. If most of the β-dependent binding events in MMS-treated cells represented active synthesis, we would expect a larger increase in the Pol IV-D103N binding lifetime.

We also explored the role of non-clamp interactions in Pol IV recruitment in MMS-treated cells. In particular, we focused on UmuD and RecA based on reports that these three proteins form a ternary complex[21, 22]. We found that MMS treatment led to the cellular reorganization of RecA, with RecA foci moving to the same midcell position as Pol IV and SSB, in agreement with a previous study that observed colocalization of Pol IV and RecA upon the induction of DNA damage with different agents[41]. This result is consistent with the generation of persistent ssDNA tracts near the replication fork by MMS damage. There was no change in Pol IV localization in a ΔumuDC strain, however. Deletion of Pol V does not affect cell survival in the presence of MMS but it does reduce the frequency of MMS-induced mutagenesis[11, 40], indicating that Pol V carries out some DNA synthesis under these conditions. Our results suggest that, although Pol V gains access to the template in MMS-treated cells, it does not compete directly with Pol IV for binding sites and that TLS by the two polymerases typically occurs in different cellular contexts. They also reveal that interactions with UmuD do not play a role in Pol IV recruitment in response to DNA damage by MMS. We note that the residual Pol IV-SSB colocalization seem for the Pol IV[R,C] mutant likely arises from interactions with other factors, possibly SSB or RecA.[20, 21]

What other interactions might be responsible for Pol IV recruitment in NFZ-treated cells if interactions with β are not required? One possible interacting partner is SSB, which was shown to bind Pol IV in a previous biochemical study[20]. However, we did not see increased Pol IV-SSB colocalization in NFZ-treated cells, except possibly at an early stage of the response, arguing that SSB is unlikely to be responsible for Pol IV recruitment under these conditions. The presence of the ε subunit of Pol III in a similar cellular position in NFZ-treated cells suggests that these may be replication intermediates, although the lack of SSB enrichment indicates that they are unlikely to represent normal replication forks. Other putative Pol IV binding partners are UmuD and RecA. As in MMS-treated cells, there was no change in Pol IV localization in a ΔumuDC strain after treatment with NFZ, indicating that UmuD is not involved in Pol IV recruitment under these conditions. We found that RecA localized to the same cellular position as Pol IV in NFZ-treated cells, suggesting a possible interaction. RecA also localized to the same position in undamaged cells, however, as previously reported[45]. Furthermore, there was a decrease in the average number of RecA foci per cell after NFZ treatment. Therefore, if RecA is responsible for recruiting Pol IV in NFZ-treated cells, it is not clear how it does so in a damage-dependent manner.

In the case of NFZ, our results demonstrate that β-independent interactions, whether with RecA or with other factors, play a role in the response of Pol IV to DNA damage. Future studies will be needed to better elucidate which non-clamp interactions are responsible for Pol IV recruitment and to determine if they stimulate TLS in cells. Radial distribution function analysis shows that Pol IV is not strongly enriched at replication forks at any point during the response to NFZ damage, in contrast to the large and stable enrichment seen in MMS-treated cells. This result suggests that TLS is carried out rapidly by single copies of Pol IV at the replication fork upon NFZ treatment, consistent with recent in vitro experiments which have demonstrated that Pol IV can exchange with Pol III and carry out rapid bypass of $N^2$-dG DNA adducts[12, 18]. In this case, the Pol IV recruitment that we observe might represent molecules involved in a different pathway that leads to greater Pol IV enrichment or occurs on a slower timescale, which would explain the longer Pol IV binding lifetime in NFZ-treated cells. In addition to its role in replication-coupled TLS, Pol IV has also been implicated in error-prone recombination through extension of D-loops[52], even in the absence of β, and transcription-coupled TLS through interactions with the transcription elongation factor NusA.[53, 54]

This study points to an unexpected degree of complexity in the mechanism of TLS polymerase recruitment in bacteria, which we show is dependent on the type of DNA damage. As a further indication of complexity, a fundamentally different recruitment mechanism has been observed for the E. coli TLS polymerase Pol V, in which Pol V is first sequestered to cell membranes and then gradually released into the cytosol[55]. Our results suggest that in addition to the differences in the initial stage of recruitment of different TLS polymerases, the type of DNA damage strongly influences whether polymerases are able to gain access to stalled replisomes or to ssDNA gaps left behind the replication fork.

## Methods

**Plasmid and bacterial strain construction.** Bacterial strains bearing fluorescent protein fusions to Pol IV and other proteins were constructed using lambda Red recombineering[23] and P1vir phage transduction (Supplementary Note 1). Detailed information about the construction of each strain (Supplementary Note 2) as well as tables of oligonucleotides, plasmids, and strains (Supplementary Tables 1–3) are provided in the Supplementary Information.

**Culture conditions and sample preparation for microscopy.** Glycerol stocks were streaked on LB plates with appropriate antibiotics and incubated overnight at

37 °C. Antibiotic concentrations used to select for chromosomal fusions were 30 μg/mL for kanamycin and 20 μg/mL for chloramphenicol. Ampicillin was used at 100 μg/mL to maintain the plasmid pASK-IBA3plus-HupA-EYFP. Single colonies were picked and used to inoculate 3 mL LB cultures which were grown for ~8 h in a roller drum at 37 °C. These overday cultures were then diluted 1:1000 into 3 mL freshly prepared M9 media and rolled overnight at 37 °C. The M9 media was supplemented with 0.4% glucose, 1 mM thiamine hydrochloride, 0.2% casamino acids, 2 mM MgSO$_4$, and 0.1 mM CaCl$_2$. The following morning, imaging cultures of 50 mL supplemented M9 glucose media were inoculated with a 1:200 dilution of the overnight cultures and grown at 37 °C shaking at ~225 r.p.m. Liquid cultures did not contain antibiotics except when needed to maintain plasmids. For full induction of the lacZ::ssb-mypet allele, IPTG was added to the overnight and imaging cultures at 0.5 mM final concentration.

Imaging cultures were grown until early exponential phase (OD$_{600nm}$ ≈ 0.15). A 1 mL aliquot was removed, pelleted by centrifugation at 8609×g, and resuspended in a few μL of the remaining supernatant. A small volume of the concentrated culture (<1 μL) was deposited on agarose pads containing M9 glucose media supplemented with 2 mM MgSO$_4$, and 0.1 mM CaCl$_2$ and sandwiched between two coverslips for imaging. Samples were imaged for up to 45 min after preparation. Agarose pads were prepared by dissolving GTG agarose (NuSieve) at 3% concentration in media and melting it on a heat block at 65 °C. A 500 μL aliquot of the molten agarose was deposited between two 25 × 25 mm coverslips (VWR), cleaned by rinsing with ethanol and deionized (DI) water, and allowed to solidify for ~15 min. The coverslip contacting the objective was cleaned by sonication to lower the background fluorescence. Coverslips were sonicated in ethanol and 1 M KOH for two cycles of 30 min each, washing with DI water in between, and then stored in DI water until use.

**NFZ and MMS and formaldehyde treatment for microscopy**. A 100 mM stock solution of nitrofurazone (NFZ) (Sigma-Aldrich) in dimethylformamide (DMF) was prepared fresh. When imaging cultures reached OD$_{600nm}$ ≈ 0.15, the NFZ stock (or an equivalent amount DMF for a solvent control) was added to the flask at a final concentration of 100 μM, corresponding to a 1:1000 dilution. A similar procedure was used for treatment with 40 μM NFZ but with a 40 mM stock solution. Cultures were grown for an additional 1 h (in most experiments) or 20 min (where indicated) and then imaged as normal.

Cells were treated with methyl methanesulfonate (MMS) following a previously published protocol[26] in which MMS was included in the agarose pads at 100 mM concentration. Cells were collected and concentrated as normal, deposited on an MMS-containing agarose pad, incubated for 20 min (in most experiments) or 1 h (where indicated) at room temperature, and then imaged on the same pad.

Cells were fixed using formaldehyde following a previously published protocol[26]. A 5 mL aliquot of culture was removed at OD$_{600nm}$ ≈ 0.15, pelleted by centrifugation at 7197×g for 5 min, and resuspended at 2× concentration in 2.5% formaldehyde in phosphate-buffered saline (PBS). Cells were incubated for 45 min at room temperature, then pelleted again and resuspended in a small volume of fresh supplemented M9 glucose medium.

**NFZ cell sensitivity and viability assays**. A plating assay was used to assess cell sensitivity to NFZ. Overnight LB cultures were diluted 1:1000 in 50 mL LB and incubated shaking at 37 °C until OD$_{600nm}$ ≈ 1.0. Aliquots were removed and serial dilutions were made in 0.9% NaCl. A dilution series was then stamped on LB agar plates containing either DMF solvent or NFZ at 8 or 10.5 μM concentrations. Plates were incubated at 37 °C and imaged after 16 h.

Cell survival after NFZ treatment under the imaging experiment conditions was assayed by collecting cultures before NFZ treatment and after 1 h treatment with 40 μM or 100 μM NFZ or DMF solvent. Serial dilutions were plated on LB agar plates and incubated at 37 °C for 16 h, and then colony-forming units (CFUs/mL) were counted.

Western blot analysis of whole cell lysates was used to verify the expression level of Pol IV-PAmCherry fusion proteins (Supplementary Figures 4a and 18). Additional detail is provided in Supplementary Note 5.

**Microscopy**. Imaging was performed on a Nikon TE2000 microscope designed for two-color single-molecule fluorescence imaging. GFP and YFP variants were excited with a 514 nm laser (Coherent Sapphire, 150 mW). PAmCherry was activated with a 405 nm laser (Coherent OBIS, 100 mW) and imaged with a 561 nm laser (Coherent Sapphire, 200 mW). The laser powers were adjusted individually using neutral density filters. The 405 nm, 514 nm, and 561 nm laser beams were expanded with a telescope, passed through excitation filters (Chroma ZET405/20X, ZET514/10X, and ZET561/10X respectively), and combined with dichroic filters (Chroma ZT405rdc, Chroma ZT514rdc, and a mirror, respectively), then expanded further by a second telescope. The beams were focused to the back focal plane (BFP) of a Nikon CFI Apo 100×/1.49 NA TIRF objective with a 400 mm focal length lens. The lens was translated using a micrometer to move the focused beam away from the center of the objective BFP in order to obtain highly inclined thin illumination, or near-TIRF.[56] RecA-GFP and HU-YFP were imaged using a single-band TIRF filter cube (Chroma TRF49905, containing a ZT514rdc dichroic filter, a ET545/40 m emission filter, and a ET525lp longpass filter). Imaging of Pol IV-

PAmCherry, ε-PAmCherry, SSB-mYPet, and ε-mYPet, either individually or in dual-color colocalization experiments, was performed using a custom dual-band TIRF filter cube (Chroma 91032 Laser TIRF Cube, containing a ZT405/514/561rpc dichroic filter, ZET442/514/561m emission filter, and ET525lp longpass filter). Images were recorded using a Hamamatsu ImageEM C9100-13 EMCCD camera.

Sample position was controlled with a motorized microscope stage and a piezo stage (Mad City Labs MicroStage and Nano-LPS100). Computer controlled shutters (Uniblitz VS14) and a filter wheel (Thorlabs FW103H) were used to automate excitation sequences for different types of imaging experiments and to ensure reproducibility. For all movies, brightfield images of cells were recorded using white light transillumination.

Short exposure single-color PALM imaging was performed using an integration time of 13.3 ms (~71 Hz, accounting for the camera readout time) with an initial pre-bleaching period of 400 frames of ~120 W/cm$^2$ 561 nm excitation. After this pre-bleaching period, continuous 405 nm and 561 nm excitation was used to activate and image PAmCherry with the same 561 nm power. The 405 nm laser power was set so that on average no more than one PAmCherry molecule was activated per cell at any time. Because the activation rate drops as PAmCherry molecules are irreversibly photobleached, the 405 nm power was increased in two steps at fixed time points during the movie (starting at ~15 mW/cm$^2$ power and increasing to ~ 105 mW/cm$^2$ power).

Long exposure two-color PALM imaging was performed using an integration time of 250 ms (~4 Hz). An initial pre-bleaching period of 50 frames was performed using ~120 W/cm$^2$ 561 nm excitation, after which the 561 nm laser power was lowered 10-fold to ~12.5 W/cm$^2$ and 10 additional frames were recorded. Next 10 frames of 514 nm excitation at ~0.4 W/cm$^2$ (for SSB-mYPet and MukB-mYPet), ~1.4 W/cm$^2$ (for ε-mYPet), or ~0.2 W/cm$^2$ (for YPet-β) power were recorded to image YFP variant fusion proteins. Finally, continuous 405 nm and 561 nm activation was used to activate and image PAmCherry. The 405 nm power was increased in two steps at fixed time points during the movie (starting at ~2.5 mW/cm$^2$ power and increasing to ~17.5 mW/cm$^2$ power). ParC-PAmCherry imaging used the same 405 nm activation power as for Pol IV. ε-PAmCherry and Pol I-PAmCherry imaging used a lower initial 405 nm activation power of ~1.4 mW/cm$^2$ and HU-PAmCherry imaging used ~0.06 mW/cm$^2$.

RecA-GFP imaging was performed using an integration time of 250 ms (~4 Hz) with continuous 514 nm excitation at ~60 mW/cm$^2$ power. HU-EYFP imaging was performed using an excitation time of 50 ms (~20 Hz) with continuous 514 nm excitation at ~120 mW/cm$^2$ power.

All imaging experiments were repeated for at least two biological replicates (imaging cultures).

**Image analysis**. Image analysis was automated using MATLAB-based packages and custom scripts. Cell outlines were obtained from brightfield images using MicrobeTracker[57]. Cells were excluded from analysis if they contained a bright spot that did not bleach during the 561 nm pre-bleaching period or if they had an unusually high fluorescence background. Crosstalk between the mYPet and PAmCherry channels under our imaging conditions was only observed for a small fraction of cells treated with 100 mM MMS for 1 h. In that case, cells that contained a track starting in the first frame of the PALM movie were excluded from analysis, since activation of PAmCherry in the first frame of 405 nm excitation was rare and these tracks appeared largely to be due to crosstalk. A very small number of cells, fewer than 1% of the total, were removed manually due to severe crosstalk that was not detected by the automated procedure. In total, ~7% of cells were removed. Dividing cells were split if a septum was visible in the brightfield image and if there was a dip in the fluorescence background intensity corresponding to the position of the septum.

PAmCherry and mYPet fusion proteins were detected, fit to symmetrical 2D Gaussian point spread functions (PSFs), and tracked using the MATLAB-based package u-track[58]. A custom script was used to generate regions of interest (ROIs) based on the cell outlines generated by MicrobeTracker. u-track analysis was then performed separately for each ROI. The point source detection algorithm was used for spot detection[59]. The free parameters for PSF fitting were x and y centroid, amplitude, width (σ), and background offset. For tracking, the nearest neighbor distance was not used to expand the search radius. Unless otherwise noted, default parameters were used for spot detection, fitting, and tracking.

For short exposure single-color PALM movies, a significance threshold of $\alpha = 10^{-2}$ was used. Gaps of one frame in length were permitted in tracks to account for PAmCherry blinking or missed localizations, but tracks were required to be at least two frames in length. A multiplier factor of 5 was used for the Brownian search radius.

For long exposure two-color PALM movies, a more stringent significance threshold of $\alpha = 10^{-6}$ was used. Gaps were not allowed in tracks, and tracks of one frame in length were permitted. The Brownian search radius upper bound was set to three pixels. Tracks were further filtered on the basis of the mean track PSF width in order to identify statically bound molecules. Mobile molecules have a broader PSF than static molecules due to motion blurring, as shown previously[26]. A mean track PSF width was calculated as the average of the individual PSF widths of each single-frame localization in the track. Static tracks were identified as those with a mean track PSF within two standard deviations of the mean value in fixed cells. The distribution of mean track PSF width in fixed cells is shown in

Supplementary Fig. 2b. From the Gaussian fit, this distribution has a mean ± s.d. of 148.3 ± 31.4 nm, corresponding to a cutoff range of mean PSF width from 85.5–211.0 nm for static tracks.

For SSB-mYPet, ε-mYPet, and YPet-β foci in long exposure two-color PALM movies, an average projection of the first 5 frames of 514 nm excitation was analyzed. A significance threshold of $\alpha = 10^{-5}$ was used. Detected spots were further filtered on the basis of the background offset value, which was required to be above the camera offset level (1500 counts). For RecA-GFP foci, an average projection of the first 50 frames of the movie was analyzed with the same approach used for mYPet foci. For HU-EYFP movies, an average projection of the first 50 frames of the movie was analyzed.

**Data analysis.** *Diffusion coefficient analysis:* apparent two-dimensional (2D) diffusion coefficients ($D^*$) were calculated for short exposure PALM trajectories as:

$$D^* = \frac{MSD}{4\Delta t}$$

where $\Delta t$ is the time interval (i.e., the reciprocal of the frame rate) and the mean-squared displacement (MSD) is:

$$MSD = \frac{1}{N-1}\sum_{i=1}^{N-1}(x_{i+1} - x_i)^2 + (y_{i+1} - y_i)^2$$

Here $N$ is the number of frames in the trajectory and $N-1$ is the number of steps. Because statistical errors in the MSD may be significant for short trajectories, only tracks with $N \geq 5$ were included in the analysis.

A threshold $D^*$ value for static tracks was determined from the apparent diffusion coefficient distribution for cells fixed with formaldehyde (Supplementary Fig. 1b). Under these conditions, 95% of tracks had $D^* \leq 0.275\ \mu m^2/s$, which was chosen as the threshold value.

*Cell rotation and normalization:* cell outlines were rotated and normalized so that cellular coordinates range from 0 to 1 along the long ($x$) axis and short ($y$) axis. Briefly, an offset value was subtracted from the cell mesh coordinates to set the coordinates of one cell pole as $x = 0$, $y = 0$. Next the rotation angle was defined as the angle formed by the line between the two cell poles and the $x$ axis. The subtracted cell mesh was rotated by this angle to align the cell poles with the $x$ axis. The rotated cell mesh was divided by the cell length and then the minimum $x$ and $y$ coordinates were set to 0. Following this normalization, the $x$ and $y$ cell coordinates range from 0 to 1. Coordinates of trajectories or foci within the cell were rotated and normalized following the same procedure. (See Supplementary Fig. 2c for an example of this process.)

*Cellular localization analysis:* cellular localization distributions in long exposure PALM imaging were generated by taking the centroid of foci or the mean trajectory $x$ and $y$ positions. This approach weights all trajectories equally regardless of track length.

*Cell length filtering:* where indicated, cellular localization distributions and HU-EYFP intensity profiles were filtered using a cell length threshold of 3.2 μm to differentiate small and large cells. This cell length threshold was determined by analyzing the cell length distributions for cells with one or two nucleoid lobes in the HU-EYFP intensity profiles. The threshold was chosen to minimize both the number of cells below the threshold with two nucleoid lobes and the number of cells above the threshold with one nucleoid lobe.

*Colocalization analysis:* single-cell Pol IV-SSB colocalization analysis was performed by taking the distance between each localization in a static Pol IV-PAmCherry track and the nearest SSB-mYPet focus in the cell. Then a mean distance was calculated for each Pol IV track by averaging the single-frame localization distances. Colocalization was also characterized using radial distribution analysis[31, 32]. The radial distribution function $g(r)$ expresses the increased likelihood of Pol IV localization at a distance $r$ from a SSB focus relative to random cellular localization. It accounts for the geometry of the cell and differences in cell size between different strains and treatment conditions. To calculate $g(r)$, the mean Pol IV-SSB distances were determined for each cell as described above. Then an equivalent number of random localizations were generated within the cell outline and the distance to the nearest SSB focus was calculated for each random localization. This procedure was repeated for each cell. Histograms were generated for the measured and random distance distributions and the measured histogram was divided by the random histogram to give $g(r)$. A $g(r)$ value of 1 indicates no enrichment relative to a random distribution. To account for variability in the simulated random localization distributions, we repeated this analysis for 100 random distributions and took the mean of the resulting 100 $g(r)$ curves. For each data set, an independent random distribution was generated in the same fashion and normalized by the 100 random distributions. These 100 random $g(r)$ curves were averaged to give a mean random $g(r)$ function. Deviations of the random $g(r)$ curve from 1 reveal variability due to the finite sample size. As another measure of variability, Supplementary Fig. 3e,f show the 100 experimental and random $g(r)$ curves for representative sample sizes. The same approach was used for colocalization analysis of other PAmCherry and mYPet fusion pairs.

We note that several factors can introduce an apparent offset between perfectly colocalized molecules in these measurements, including localization error for both PAmCherry and mYPet, the fact that SSB-mYPet foci are larger than diffraction limited, chromatic effects, and possible chromosome motion or stage drift over the course of the PALM movie.

We estimated the PAmCherry localization precision $\sigma_{xy}$ under our long exposure imaging conditions as approximately 17 nm by measurements of Pol IV-PAmCherry in fixed cells. To determine this error, we calculated the offset of each track localization from the mean track position, aggregated the $x$ and $y$ offset values across all tracks, and fit the resulting distributions to Gaussian functions to determine the $x$ and $y$ localization precisions $\sigma_x$ and $\sigma_y$. Finally, the lateral localization precision $\sigma_{xy}$ was calculated as[60]:

$$\sigma_{xy} = \left(\sigma_x^2 + \sigma_y^2\right)^{1/2}$$

To check for chromatic effects in our optical configuration that might introduce an apparent shift between the PAmCherry and mYPet channels, we measured the position of TetraSpeck beads (Molecular Probes #T-7279) excited with 514 nm and 561 nm illumination and compared the position of the bead centroids. We found an apparent offset of less than 10 nm using this approach.

*Replisome focus intensity analysis:* the fluorescence intensities of SSB-mYPet, ε-mYPet, and YPet-β foci were obtained as the integrated areas of the symmetrical 2D Gaussian fits performed in u-track. We note that this analysis of focus intensities is limited by the dynamic range of the camera used in the measurement. Under our imaging conditions, approximately 14 and 7% of SSB foci in NFZ- and MMS-treated cells respectively contained at least one saturated pixel, as did 5% of β foci in MMS-treated cells. Saturation of pixels will introduce some inaccuracy into the resulting Gaussian fit. Only 3% or fewer of SSB foci in undamaged cells, β foci in undamaged or NFZ-treated cells, or ε foci under any condition contained saturated pixels.

*Nucleoid profile analysis:* average HU-EYFP intensity profiles were generated to assess the morphology and extent of the nucleoid. First, an average projection of the first 50 frames of the movie was generated. Next, the centerline of the cell was determined based on the cell outline coordinates. The fluorescence intensity along this centerline was obtained by rounding the $x$ and $y$ coordinates of each centerline point to the nearest integer and taking the intensity of that pixel in the projected image. Single-cell intensity profiles were then normalized so that an average intensity profile could be calculated across all cells. First, each single-cell intensity profile was normalized to the distance along the cell centerline by dividing the distance coordinate by the cell length. These normalized profiles were then interpolated to get intensities at standard distances along the cell length. Finally, the resulting interpolated single-cell profiles were averaged to give the mean intensity profile across all cells.

*Statistical analysis:* distributions were compared using a two-sided Wilcoxon rank-sum test with the MATLAB function rank-sum. Statistical significance was defined as $p < 0.05$.

**Data availability.** The data sets and custom-written computer code from the current study are available from the corresponding author on reasonable request.

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

## Acknowledgements

We thank Johannes Walter for providing the anti-FLAG antibody, Ann Hochschild for providing the anti-RpoA antibody, Steven Sandler for providing the RecA-GFP strain, James Weisshaar for providing the HU-EYFP plasmid, Xiaowei Zhuang for providing the mMaple3 plasmid and HU-PAmCherry allele, Rodrigo Reyes-Lamothe for providing the YPet-β allele, and David Sherratt for providing the ParC-PAmCherry MukB-mYPet strain. We particularly thank Stephan Uphoff for providing the Pol I-PAmCherry and ε-PAmCherry strains and for advice on PALM imaging. We also thank Jennifer Waters and Talley Lambert at the Harvard Medical School Nikon Imaging Center for assistance with collection of preliminary data and for other advice, and Hunter Elliott and Joy Yichao Xu at the Harvard Medical School Image and Data Analysis Core for assistance with analysis code. This work was funded by National Institutes of Health grants R01 GM114065 (to J.J.L.) and F32 GM113516 (to E.S.T.) and by a National Science Foundation Graduate Research Fellowship DGE-1144152 (to J.E.K.).

## Author contributions

E.S.T., J.E.K., and J.J.L. designed experiments. E.S.T. and J.E.K. performed experiments. E.S.T., J.E.K., and S.C. contributed new reagents. E.S.T., J.E.K., and J.J.L. analyzed data and wrote the manuscript.

## Additional information

**Competing interests:** The authors declare no competing financial interests.

