## [Peer Review File · Nature Communications]

Reviewers' Comments:

Reviewer #1:

Remarks to the Author:

This paper deals with live imaging analysis of a TLS DNA polymerase, Pol IV in *Escherichia coli*. Using a sophisticated imaging method, the authors revealed how Pol IV molecules distribute in normally growing cells and behave when cells are treated with DNA damaging agents. They showed that about 21% of Pol IV molecules were static and formed multiple foci that distribute throughout the cell. In cells treated with NFZ, the fraction of static molecules was increased to 47%, the number of foci was also doubled, and the distribution of foci was shifted toward the periphery of nucleoid and the cell pole. The foci formation and distribution in normally growing cells and NFZ-treated cells were beta-clamp independent and no relevant to replication fork. On the other hand, the behavior of Pol IV was rather different when cells were treated with MMS. The foci of Pol IV localized at mid-cell in MMS-treated cells. This change of localization was associated with the localization of SSB and required the beta-clamp interaction, suggesting an accumulation of Pol IV at stalled replication forks.

These findings are mostly unexpected and important for understanding how Pol IV molecules are recruited at DNA damage sites or stalled replication forks in cells. Experiments were carefully designed, and sufficient data from various control experiments were provided. The paper is well organized and written carefully with a detailed description of methodology and data analysis. Therefore, the reviewer has no objection to the contents of this manuscript.

Only a point the reviewer once considered to be asked is the behavior of Pol IV in a *recA* deficient cells with and without DNA damaging agents. However, as the authors mentioned, extreme sensitivities of such cells to NFZ and MMS would make the live imaging analysis difficult. Perhaps, some indirect effects of the absence of *recA* would not be ruled out.

Reviewer #2:

Remarks to the Author:

This manuscript describes single-molecule fluorescence imaging analysis of DNA polymerase IV (Pol IV) in *E. coli*. The authors examine the localization of Pol IV in cells treated with DNA damaging agents NFZ and MMS, and measure colocalization between Pol IV and SSB foci. The work is (mostly) technically proficient, but provides a relatively minor advance in our knowledge of Pol IV regulation in cells. The observation with the most potential impact – the lack of tight colocalization between Pol IV and replication forks – has not been satisfactorily demonstrated.

Major issues:

1) Cells are incubated with DNA damaging agents for a certain period of time prior to imaging – for MMS this is 20 min, for NFZ this is 60 min. Are these time-points of particular interest, or were they chosen arbitrarily? If Pol IV were to act at replication forks that become stalled at DNA damage, as described in most models of TLS, would it not be of most interest to look immediately after damage is induced? Additionally, one could imagine that the behavior of Pol IV might change dramatically as a function of time after damage induction, even in an SOS-constitutive Δ lexA background. The authors describe differences in Pol IV behavior in cells treated with MMS vs NFZ, however the fact that the measurements were performed at different time-points after treatment makes this comparison less valid for the reasons outlined above.

2) The authors describe a new localization pattern for Pol IV in cells treated with NFZ, but there is little discussion of what the unusual features near the ends of the cell correspond to. They show that the epsilon subunit, like Pol IV, is enriched near the cell poles, but that SSB is not. So either the region is occupied by replication forks that do not contain SSB, or the region is occupied by

something else, to which Pol IV and Pol III bind. In any case, the unusual localization pattern in NFZ-treated cells deserves further explanation.

3) Based on previous observations, I would have expected that both NFZ and MMS would induce significant filamentation in MG1655 backgrounds, even when *sulA* is non-functional. The authors of this manuscript, however, observe that cells become smaller upon treatment, not larger. This observation, coupled with the fact that the NFZ concentrations used are way higher than those in which cells survive treatment, make me suspect that the authors might even be examining dead cells. This could perhaps explain the unusual localization of Pol IV they observe, as well as the relatively large proportion of static foci they observe in the absence of damage. Notably, previous single-molecule imaging work from the Foster lab (which is poorly cited in this manuscript) states that they observed no static Pol IV foci in the absence of damage. In general, the authors need to show much more raw data, i.e. microscope images. It is impossible to assess what state their cells might be in during these measurements based on the current figures.

4) There is a large difference in the lifetime of foci in undamaged/MMS-treated cells versus NFZ-treated/fixed cells. The origins and implications of this difference are not discussed.

5) The idea that Pol IV rarely colocalizes with replication forks would have been the most important observation to come out of this work – but it has not been convincingly demonstrated. The histograms of colocalization distances for Pol IV and SSB (Fig 3) all show few instances of colocalisation distances < 100 nm. Presumably, this dip at zero arises from the fact that colocalization is a radial measurement – distance shells further from the measurement point have a larger area than shells close to the measurement point. In other words, the propensity for chance colocalization between the Pol IV and SSB increases with the square of the distance. This is not discussed in the manuscript. The apparent lack of short-distance colocalization events is, however, presented as evidence for Pol IV acting behind the fork, on lesions skipped over by replisomes. It's possible that this is true, but to make that statement, one would first have to determine how to properly interpret histograms of colocalization differences. It may be helpful to measure these differences for a pair of probes that are known to colocalize.

We thank the reviewers for their careful consideration of our manuscript. We have added several new experiments and analyses to address their concerns, and we believe that these revisions have significantly strengthened the manuscript. We address detailed reviewer comments below. Our responses are in blue and new manuscript text is indicated in italics.

We have tracked all major changes made in response to reviewer comments in the main text, although we did not highlight smaller changes. Changes to the Supplementary Information other than the addition of new figures were minor and were not tracked.

Reviewers' comments:

Reviewer #1 (Remarks to the Author):

This paper deals with live imaging analysis of a TLS DNA polymerase, Pol IV in *Escherichia coli*. Using a sophisticated imaging method, the authors revealed how Pol IV molecules distribute in normally growing cells and behave when cells are treated with DNA damaging agents. They showed that about 21% of Pol IV molecules were static and formed multiple foci that distribute throughout the cell. In cells treated with NFZ, the fraction of static molecules was increased to 47%, the number of foci was also doubled, and the distribution of foci was shifted toward the periphery of nucleoid and the cell pole. The foci formation and distribution in normally growing cells and NFZ-treated cells were beta-clamp independent and no relevant to replication fork. On the other hand, the behavior of Pol IV was rather different when cells were treated with MMS. The foci of Pol IV localized at mid-cell in MMS-treated cells. This change of localization was associated with the localization of SSB and required the beta-clamp interaction, suggesting an accumulation of Pol IV at stalled replication forks.

These findings are mostly unexpected and important for understanding how Pol IV molecules are recruited at DNA damage sites or stalled replication forks in cells. Experiments were carefully designed, and sufficient data from various control experiments were provided. The paper is well organized and written carefully with a detailed description of methodology and data analysis. Therefore, the reviewer has no objection to the contents of this manuscript.

Only a point the reviewer once considered to be asked is the behavior of Pol IV in a *recA* deficient cells with and without DNA damaging agents. However, as the authors mentioned, extreme sensitivities of such cells to NFZ and MMS would make the live imaging analysis difficult. Perhaps, some indirect effects of the absence of *recA* would not be ruled out.

We agree that the role of RecA in the response of Pol IV to DNA damage is quite interesting and worthy of further study, although the sensitivity of *recA*⁻ cells might confound the interpretation of imaging experiments. In response to the reviewer's comment, however, we did attempt to

knock out *recA* in our imaging strain using two methods: 1) lambda Red recombineering and 2) P1*vir* phage transduction. We successfully constructed a $\Delta recA::cat$ strain in the wild-type (WT) MG1655 background using recombineering. We were then able to generate a P1*vir* phage stock from this strain (after transformation of a plasmid expressing *recA*, because P1 replication is inhibited in *recA*⁻ cells) and to transfer the $\Delta recA::cat$ allele to a separate MG1655 isolate using P1*vir* transduction. However, we were unable to move this $\Delta recA::cat$ allele into our imaging strain using either of these approaches for reasons that are unclear. Future work will be needed to explore the role of RecA further.

Reviewer #2 (Remarks to the Author):

This manuscript describes single-molecule fluorescence imaging analysis of DNA polymerase IV (Pol IV) in *E. coli*. The authors examine the localization of Pol IV in cells treated with DNA damaging agents NFZ and MMS, and measure colocalization between Pol IV and SSB foci. The work is (mostly) technically proficient, but provides a relatively minor advance in our knowledge of Pol IV regulation in cells. The observation with the most potential impact – the lack of tight colocalization between Pol IV and replication forks – has not been satisfactorily demonstrated.

Major issues:

1) Cells are incubated with DNA damaging agents for a certain period of time prior to imaging – for MMS this is 20 min, for NFZ this is 60 min. Are these time-points of particular interest, or were they chosen arbitrarily? If Pol IV were to act at replication forks that become stalled at DNA damage, as described in most models of TLS, would it not be of most interest to look immediately after damage is induced? Additionally, one could imagine that the behavior of Pol IV might change dramatically as a function of time after damage induction, even in an SOS-constitutive $\Delta lexA$ background. The authors describe differences in Pol IV behavior in cells treated with MMS vs NFZ, however the fact that the measurements were performed at different time-points after treatment makes this comparison less valid for the reasons outlined above.

The 20 min time point for MMS treatment was chosen based on a previous study that looked at Pol I recruitment in response to DNA damage¹. Unlike MMS, which directly alkylates DNA, NFZ is activated after processing by cellular nitroreductases.^{2,3} Therefore we initially used a longer 1 h incubation for NFZ to allow DNA damage to accumulate. We do, however, agree that our manuscript would benefit from a more direct comparison between NFZ and MMS in terms of exposure time, as well as the additional time points for each damaging agent. In response, we have now characterized the cellular localization of Pol IV after 20 min and 2 h exposure to 100 μ M NFZ and 1 h exposure to 100 mM MMS. These new data are shown in Supplementary Fig. 8. In the case of NFZ, the characteristic Pol IV localization pattern is already present after 20 min, although less pronounced. There is no mid-cell localization, as seen for MMS, at this earlier

time point. Pol IV localization after 2 h incubation with NFZ is qualitatively similar to the 1 h condition, indicating that the response is stable over time. Likewise, we see strong mid-cell localization of Pol IV after 1 h MMS treatment, consistent with the results for 20 min treatment. Overall these new data confirm that the distinct Pol IV localization patterns for NFZ and MMS treatment are not simply due to differences in time after the induction of DNA damage. Although it would be interesting to look at exposure times even shorter than 20 min, preparing the sample and recording PALM movies takes a certain amount of time, and thus we don't feel that we have adequate time resolution to reliably measure the behavior immediately after induction of damage.

2) The authors describe a new localization pattern for Pol IV in cells treated with NFZ, but there is little discussion of what the unusual features near the ends of the cell correspond to. They show that the epsilon subunit, like Pol IV, is enriched near the cell poles, but that SSB is not. So either the region is occupied by replication forks that do not contain SSB, or the region is occupied by something else, to which Pol IV and Pol III bind. In any case, the unusual localization pattern in NFZ-treated cells deserves further explanation.

We have carried out a number of experiments attempting to understand the Pol IV localization pattern in NFZ-treated cells. In particular, we show that the cellular localization of RecA is similar to that of Pol IV, although (as described in our response to Reviewer #1) we have been unable to test for direct recruitment by deletion of *recA* in our imaging strain. Further work will be needed to characterize the Pol IV-RecA interaction and to determine whether it plays a role in recruitment, although we believe that the complexity of RecA's involvement in multiple cellular and DNA damage response pathways, as well the extreme DNA damage sensitivity of *recA*⁻ strains, places such work outside the scope of this study. Although we do not find strong enrichment of SSB in this cellular position, we note in the text that measurements of the centroid of SSB-mYPet foci do not represent the full cellular distribution of SSB, and there is likely some overlap between the Pol IV localization and the periphery of the SSB-mYPet foci. Thus although there is not major enrichment of SSB in the region occupied by Pol IV, some molecules may still be present. In our response to point #3 below, we discuss other experiments and analysis that argue for the physiological relevance of this localization. While the exact molecular mechanism of this recruitment remains elusive, by characterization of several Pol IV mutants we have demonstrated that interactions with the β clamp are not required and that the N-terminal catalytic domain of Pol IV is sufficient for localization.

We have added the following sentence to the third-to-last paragraph of the discussion section to clarify what we can state about the localization pattern on the basis of our current experiments:

The presence of the ϵ subunit of Pol III in a similar cellular position in NFZ-treated cells suggests that these may be replication intermediates, although the lack of SSB enrichment indicates that they are unlikely to represent normal replication forks.

It also possible that the localization of Pol IV reflects its involvement in other pathways, such as D-loop extension or transcription-coupled TLS, as described in the second-to-last paragraph of the discussion section.

3) Based on previous observations, I would have expected that both NFZ and MMS would induce significant filamentation in MG1655 backgrounds, even when *sulA* is non-functional. The authors of this manuscript, however, observe that cells become smaller upon treatment, not larger. This observation, coupled with the fact that the NFZ concentrations used are way higher than those in which cells survive treatment, make me suspect that the authors might even be examining dead cells. This could perhaps explain the unusual localization of Pol IV they observe, as well as the relatively large proportion of static foci they observe in the absence of damage. Notably, previous single-molecule imaging work from the Foster lab (which is poorly cited in this manuscript) states that they observed no static Pol IV foci in the absence of damage. In general, the authors need to show much more raw data, i.e. microscope images. It is impossible to assess what state their cells might be in during these measurements based on the current figures.

Although significantly lower NFZ concentrations are used in plate-based sensitivity assays, which entail long exposures to the drug over many hours and generations, these assays are not directly comparable to relatively short treatments in liquid culture (for another example, please compare Fig. 2 and S1 of Uphoff et al. PNAS 2013). Previously we characterized cell survival after 1 h treatment with 100 μ M NFZ in liquid culture. We did find a moderate reduction in viable cell counts under these conditions, but not as great as would be observed for treatment with the same concentration on a plate. We have now also measured cell survival after 1 h treatment with 40 μ M NFZ in liquid culture. Under this condition, we find no loss of viable cell counts. Cell growth is slowed in comparison to undamaged cells, but there is still a modest increase in the number of viable cells after 1 h of NFZ incubation (Supplementary Note 4). The presence of an increased fraction of static Pol IV molecules and their characteristic localization pattern in the 40 μ M NFZ condition (see new data in Fig. 4c) therefore suggests that this effect is not due to dead cells.

To confirm that the observed Pol IV localization in the 100 μ M NFZ condition does not arise from a small population of potentially non-viable cells with a large number of static tracks, we have added a new supplemental figure, Supplementary Fig. 7, in which we compared the cellular localization of Pol IV in two populations sorted by the number of static tracks per cell. The Pol IV localization pattern is qualitatively the same for cells in the top 25% (most static Pol IV

tracks) and the bottom 75%. We find similar results for other percentile cutoffs (data not shown), indicating that the localization pattern is consistent across the population of cells, and strongly suggesting that changes in Pol IV behavior are not just seen in a minority of dead cells.

In response to the reviewer's concerns about the state of the cells after DNA damage, we have added a new supplemental figure, Supplementary Fig. 5, showing brightfield micrographs of cells in three randomly-selected fields of view for each of the main treatment conditions in the paper: undamaged, 40 μ M NFZ (1 h), 100 μ M NFZ (1 h), and 100 mM MMS (20 min). We have also included distributions of the cell length measured under these conditions. Regarding the reviewer's point about cell size, our interpretation of the data is that the cell size in our experiments is dictated by a number of factors including the growth rate and the inhibition of filamentation in the *sulA*⁻ background. Suppression of cell growth by DNA damage might be expected to lead to a reduction in cell size, which depends on the growth rate for *E. coli* and other bacterial species⁴. We do not believe that the lack of filamentation means that cells are not viable; for example, we observe a very small increase in cell length (approximately 8%) for cells treated with 40 μ M NFZ in comparison to undamaged cells, although we find no loss of viable cells under those conditions. In comparison to our *sulA*⁻ imaging strain, we observed relatively modest increases in cell length (approximately 25 – 50%) for *sulA*⁺ MG1655 strains treated with 40 μ M NFZ (data not shown). These results are consistent with a previous report that NFZ induces the SOS response relatively weakly⁵, which we have reproduced in our own laboratory (data not shown). In the case of MMS, a previous study used the same treatment condition for strains in the *sulA*⁺ background AB1157¹; although the cell size was not quantified, micrographs in that study do not show significant filamentation after treatment with 100 mM MMS for 20 min.

Regarding the presence of static Pol IV tracks in the absence of damage, we note that our PALM imaging approach is distinct from and more sensitive than the imaging employed in the Foster manuscript. The Pol IV foci observed in the Foster lab's paper contain multiple copies of Pol IV and are likely fairly long-lived, whereas our PALM imaging is sensitive to relatively transient binding events of single molecules. It is unsurprising that the Foster lab imaging conditions are unable to resolve the short binding events of single Pol IV molecules that we observe in undamaged cells, despite being able to resolve the longer binding events and greater Pol IV enrichment in damaged cells.

We also agree that our manuscript should include more discussion of how our results relate to the findings of the Foster lab's paper. We have added the following sentences (new text in *italics*) in the discussion section to address this previous work and to put our study in context:

Consistent with studies of other DNA and RNA polymerases, we observed two populations of Pol IV in cells, one diffusing and one statically bound. Selective imaging

of static molecules revealed that Pol IV binds randomly throughout the cell under normal growth conditions. *A previous study, using a Pol IV-EYFP construct overexpressed from a plasmid, did not observe Pol IV binding in undamaged cells; unlike PALM, however, this imaging technique is not sensitive to short binding events of single molecules.*

A prior study reported the formation of multi-copy Pol IV foci in response to treatment with the DNA damaging agent 4-nitroquinoline 1-oxide (4-NQO) or double-strand break induction. In this work, we found that DNA damage by NFZ or MMS led to strong Pol IV enrichment at particular cellular positions, with the nature of the enrichment dependent on the type of damage.

We found that MMS treatment led to the cellular reorganization of RecA, with RecA foci moving to the same midcell position as Pol IV and SSB, *in agreement with a previous study that observed colocalization of Pol IV and RecA upon the induction of DNA damage with different agents.*

We also wish to note several ways in which our paper differs from and goes beyond the Foster lab's paper. In addition to the use of a chromosomally encoded Pol IV fusion and more sensitive PALM imaging, our work also provides quantitative information regarding the cellular localization, enrichment, and binding dynamics of Pol IV. By a quantitative measurement of colocalization, we demonstrate recruitment of Pol IV to replication forks upon DNA damage. Additionally, we show for the first time that the nature of the DNA damage affects Pol IV response and we explore the role of the important Pol IV- β clamp interaction. We find both surprising β -independent recruitment of Pol IV in the case of NFZ treatment and evidence for β -dependent recruitment of Pol IV to clamps behind the replication fork in the case of MMS treatment.

4) There is a large difference in the lifetime of foci in undamaged/MMS-treated cells versus NFZ-treated/fixed cells. The origins and implications of this difference are not discussed.

The different Pol IV recruitment mechanism in NFZ- and MMS-treated cells likely explains the different binding lifetimes as well as the localization patterns. We have now expanded and clarified the discussion of the Pol IV binding lifetime in the text. In the first paragraph of the discussion section, we interpret the absence of strong Pol IV enrichment or long-lived binding events near the replication fork in undamaged cells as supporting the model that the access of Pol IV to the DNA template is restricted in the absence of DNA damage. We have added the following text to the third paragraph of the discussion to describe one possible interpretation of the short binding lifetime in MMS-treated cells:

In contrast to the increased Pol IV binding lifetime in NFZ-treated cells, the slight decrease in lifetime upon MMS treatment shows either that many of the localizations that we observe represent transient clamp-binding without TLS or that Pol IV does not carry out extensive synthesis under these conditions.

In the sixth paragraph of the discussion, we discuss other pathways in which Pol IV appears to play a role, including D-loop extension and transcription-coupled TLS. It is possible that Pol IV dynamics in these pathways are slower than in replication-coupled TLS, but to our knowledge there have been no direct measurements addressing this question. Thus the longer binding lifetime in NFZ-treated cells might reflect the involvement of Pol IV in a different process. We have modified the following sentence in the second-to-last paragraph of the discussion section to address this possibility:

In this case, the Pol IV recruitment that we observe might represent molecules involved in a different pathway that leads to greater Pol IV enrichment or occurs on a slower timescale, which would explain the longer Pol IV binding lifetime in NFZ-treated cells.

5) The idea that Pol IV rarely colocalizes with replication forks would have been the most important observation to come out of this work – but it has not been convincingly demonstrated. The histograms of colocalization distances for Pol IV and SSB (Fig 3) all show few instances of colocalization distances < 100 nm. Presumably, this dip at zero arises from the fact that colocalization is a radial measurement – distance shells further from the measurement point have a larger area than shells close to the measurement point. In other words, the propensity for chance colocalization between the Pol IV and SSB increases with the square of the distance. This is not discussed in the manuscript. The apparent lack of short-distance colocalization events is, however, presented as evidence for Pol IV acting behind the fork, on lesions skipped over by replisomes. It's possible that this is true, but to make that statement, one would first have to determine how to properly interpret histograms of colocalization differences. It may be helpful to measure these differences for a pair of probes that are known to colocalize.

We have attempted to clarify our discussion of the colocalization analysis, which was not clear in the original manuscript. We do not interpret the dip at zero in the colocalization plots as being evidence for Pol IV acting behind the replication fork; that argument is based on our estimate of Pol IV copy number near the fork (see Supplementary Note 6) and the steady-state copy number of the β clamp near the fork under normal growth conditions. Based on the large number of Pol IV molecules that we estimate to bind in the region of the replication fork, we argue that it's unlikely that they are binding just to the two copies of β that are directly coupled to the replisome. Instead, since the observed recruitment is dependent on β , we are arguing that this magnitude of Pol IV enrichment suggests that Pol IV is interacting with β behind the replication fork.

Instead, the dip in Pol IV-SSB separation distance at zero is the result of multiple factors that can create an apparent offset: the localization error of PAmCherry and mYPet, the fact that SSB-mYPet foci are larger than diffraction limited, chromatic effects, and possible chromosome motion or stage drift over the course of the PALM movie. By imaging Pol IV-PAmCherry in fixed cells, we can estimate the Pol IV localization error σ_{xy} as approximately 17 nm under our imaging conditions. By measurements of multi-color TetraSpeck beads under 514 nm and 561 nm excitation, we estimate that chromatic effects are minimal in our system, leading to offsets of less than 10 nm between the channels. The other factors are harder to quantify. The reviewer's point about colocalization being a radial measurement is also valid. For this reason, studies using similar approaches frequently treat offsets of less than 200 nm as representing colocalized molecules^{6,7}.

We have added the following sentence to the text (in the first paragraph of the section entitled "MMS treatment alters the localization of static Pol IV") to clarify these points:

There was a peak in the Pol IV-SSB distance distribution at approximately 115 nm in addition to a broad shoulder around 500 nm; as the SSB foci are larger than diffraction-limited PSFs, this 115 nm offset is well within the average focus width, indicating colocalization of Pol IV and SSB. The dip at Pol IV-SSB separation distances close to zero reflects a number of factors, including the localization error for PAmCherry and mYPet (see Methods), and the measured separation cannot therefore be interpreted as the actual distance of Pol IV from the center of replication forks.

We have also added a more detailed discussion of this point in the Methods section (under the header "Pol IV-SSB colocalization analysis") including analysis of the PAmCherry localization error and a characterization of chromatic effects.

Finally, we note that a simple Monte Carlo simulation incorporating PAmCherry and mYPet localization error alone can qualitatively reproduce the observed behavior, although the magnitude of the dip and the displacement from zero are sensitive to the parameters chosen. Briefly, we assume perfect PAmCherry-mYPet colocalization but with localization errors of approximately 20 nm for both fluorophores. For each simulated distance measurement, we generate a mYPet focus position centered at the cell midpoint with the addition of a randomly generated offset in x and y sampled from a Gaussian distribution representing the localization error. Then we generate a PAmCherry position, centered at the same point but again sampling from a Gaussian distribution representing the localization error. We calculate the distance between these two points and repeat this procedure a large number of times ($n = 1,000$). A histogram of the resulting apparent PAmCherry-mYPet distances shows a dip at short separation:

References cited:

1. Uphoff, S., Reyes-Lamothe, R., Garza de Leon, F., Sherratt, D. J. & Kapanidis, A. N. Single-molecule DNA repair in live bacteria. *Proc. Natl. Acad. Sci. U. S. A.* **110**, 8063–8 (2013).
2. McCalla, D. R., Reuvers, A. & Kaiser, C. Mode of action of nitrofurazone. *J. Bacteriol.* **104**, 1126–34 (1970).
3. McCalla, D. R., Kaiser, C. & Green, M. H. Genetics of nitrofurazone resistance in *Escherichia coli*. *J. Bacteriol.* **133**, 10–6 (1978).
4. Amir, A. Is cell size a spandrel? *Elife* **6**, e22186 (2017).
5. Benson, R. W., Norton, M. D., Lin, I., Du Comb, W. S. & Godoy, V. G. An active site aromatic triad in *Escherichia coli* DNA Pol IV coordinates cell survival and mutagenesis in different DNA damaging agents. *PLoS One* **6**, e19944 (2011).
6. Zawadzki, P. *et al.* The localization and action of Topoisomerase IV in *Escherichia coli* chromosome segregation is coordinated by the SMC complex, MukBEF. *Cell Rep.* **13**, 2587–96 (2015).
7. Garza de Leon, F., Sellars, L., Stracy, M., Busby, S. J. W. & Kapanidis, A. N. Tracking low-copy transcription factors in living bacteria: the case of the lac repressor. *Biophys. J.* **112**, 1316–1327 (2017).

Reviewers' Comments:

Reviewer #2:

Remarks to the Author:

The authors have clearly put a lot of effort into improving the technical aspects of the study, however the interpretation of the results remains unsatisfactory and their main claims remain unsubstantiated. As such, my view is that the manuscript remains unsuitable for publication. The data collected by the authors are inappropriate for answering the types of biological questions that they (and the field) raise. Their attempts to find biological meaning in the data have led to severe over-interpretation (see REV2-2 and REV2-5).

Below I address each of the individual points raised in the first round of review and the corresponding replies by the authors.

REV1-1

Reviewer #1 suggested that the authors try deleting *recA*. The authors reply that they tried to make the strain and it didn't work. Fair enough, although it is a little bit surprising that they could delete *recA* from wild type cells, but not from the imaging strain. The implication would be that the labelled *pol IV* allele is interfering in the biology somehow.

REV2-1

Reviewer #2 pointed out that the time-points analysed after MMS and NFZ treatments were different. The authors have since added extra time-points for the localisation analysis, which is good. Extra time-points for the colocalisation analysis against SSB foci have not been included.

REV2-2

This relates to the new localisation pattern for *pol IV* that appears after NFZ treatment. The authors discuss this further, but the discussion adds little to the picture. The localisation pattern they observe does not depend on interaction with beta-clamps or the catalytic activity of *pol IV*. The simplest interpretation is that observed localisation pattern doesn't tell us anything about the function of *pol IV*. The authors certainly don't provide any data to link this localisation pattern with biological function, such as NFZ survival assays. What does recruitment mean if it is not a step that precedes activity? What is presented in this manuscript could equally be described as "sticking to stuff". Yes, the localisations of *RecA-GFP* and *DnaQ-PAmCherry* show some similarities to the *pol IV-PAmCherry* pattern, but this may also be coincidence. Without data that link function to localisation, no meaning can be extracted from the localisation pattern.

REV2-3

Reviewer #2 raised concerns about cell size and viability upon treatment with MMS and NFZ. The changes made to the manuscript adequately address these concerns. The cells are too sick to filament, but this clearly does not affect their viability once removed from MMS/NFZ.

REV2-4

This relates to the lifetimes of *pol IV* foci. The authors' interpretation is that most MMS-induced foci are due to non-TLS events or events that do not involve extensive synthesis (presumably TLS events with short processivity). No attempt was made to test either of these ideas using mutants impaired for beta-binding or catalytic activity. Again, without experimental links to biological function, the authors are speculating. The NFZ-induced are long-lived. Does this require catalytic activity? Or binding to beta-clamps? The authors speculate that the long-lived binding might reflect involvement in pathways such as recombination and transcription-coupled TLS. Once again the authors' data do not actually teach us anything about the biological function of *pol IV*.

REV2-5

This point was originally about the analysis of colocalisation. The authors now bring in the issue of the number of *pol IV* molecules within enrichment sites. In doing so they raise an even more

important point relating to over-interpretations of data.

I'll deal with the colocalisation distances first. Yes the authors explain that the dip at zero arises from several factors, but what remains missing is what the shapes of the plots in Figure 3 actually mean. What would a plot of colocalisation distances look like for something that is tightly colocalised? And by that I mean something MEASURED under similar circumstances to the pol IV-SSB data, i.e. same pair of fluorophores, same imaging conditions etc. What about a plot for two things that are not colocalised? What about a pair of proteins that both localise to nucleoid, but do not otherwise interact with each other? Without these kinds of calibration points, it is impossible to interpret the pol IV-SSB data.

Now the number of pol IV molecules at enrichment sites. The authors' claims are predicated on terrible use of extrapolation. The authors measure the number of static foci in their region-of-interest (fine), and then compare this to the number of static foci in fixed cells (not fine). Based on experience I think fixing is likely to severely reduce the number of fluorophores that are activateable within the cells. The authors have provided no evidence to the contrary. Next the authors' compare what they claim to be the percentage of molecules that form static foci in the ROI to the literature value of 2000 pol IV molecules per cell – really, really not OK! Firstly, this number was derived from Western blot, which are far from precise and have in many cases been shown to be inaccurate. Secondly, the authors have not compared the number of pol IV-PAmCherry molecules in the imaging strain with the number of pol IV molecules in wild type cells. Put together, these issues could impact the authors' estimates for the number of molecules in the ROI by two or three orders of magnitude! This estimate is an even worse basis for their claim that pol IV acts at gaps behind the fork than the colocalisation measurements! In their rebuttal the authors claim that the Foster group's study used a less sensitive measurement than theirs. They certainly would have noticed foci containing 150 dinB-eYFP molecules!

We thank the reviewer for careful consideration of our manuscript. We have added new experiments and analyses to address the concerns below, which we think further strengthen several aspects of our study. We address detailed reviewer comments below. Our responses are in blue and new manuscript text is indicated in italics.

We have highlighted major changes made in response to reviewer comments in the main text, although we did not highlight smaller changes. We did not highlight changes in the Supplementary Information, as they were minor other than modifications to Supplementary Notes 6 and 7 and the addition of new figures.

Reviewers' comments:

Reviewer #2 (Remarks to the Author):

The authors have clearly put a lot of effort into improving the technical aspects of the study, however the interpretation of the results remains unsatisfactory and their main claims remain unsubstantiated. As such, my view is that the manuscript remains unsuitable for publication. The data collected by the authors are inappropriate for answering the types of biological questions that they (and the field) raise. Their attempts to find biological meaning in the data have led to severe over-interpretation (see REV2-2 and REV2-5).

Below I address each of the individual points raised in the first round of review and the corresponding replies by the authors.

REV1-1

Reviewer #1 suggested that the authors try deleting *recA*. The authors reply that they tried to make the strain and it didn't work. Fair enough, although it is a little bit surprising that they could delete *recA* from wild type cells, but not from the imaging stain. The implication would be that the labelled pol IV allele is interfering in the biology somehow.

We note that the inability to delete *recA* could be due to other differences in the *lexA51* strain background relative to the wild-type MG1655 strain, and not simply the presence of the Pol IV fusion. The sensitivity and survival assays shown in the paper demonstrate that the fusion protein has wild type (WT) or near-WT TLS activity and retains the ability to interact with the β clamp (Fig. 1b and Supplementary Fig. 1a). We also reiterate that given the dramatic NFZ and MMS sensitivity of *recA* deletion strains, as Reviewer #1 acknowledged, this approach is not the best way to further explore potential RecA-dependent recruitment of Pol IV. We believe this is an important topic but, in light of the involvement of RecA in virtually all mechanisms of TLS and DNA repair, is beyond the scope of this manuscript.

REV2-1

Reviewer #2 pointed out that the time-points analysed after MMS and NFZ treatments were different. The authors have since added extra time-points for the localisation analysis, which is good. Extra time-points for the colocalisation analysis against SSB foci have not been included.

We thank the reviewer for suggesting this addition. We have now added colocalization analysis for the additional NFZ and MMS time points in Fig. 3 and Supplementary Fig. 9. In these figures, we show both the mean Pol IV-SSB distance and a new, more quantitative radial distribution function analysis (please see our response to REV2-5 for more details on this analysis). In the case of MMS, there is a small increase in colocalization for the 1 h incubation in comparison to the 20 min incubation. In the case of NFZ, there is no Pol IV-SSB colocalization for the 1 h and 2 h incubations. There is modest, β -dependent Pol IV-SSB colocalization for the 20 min incubation, which is comparable to or slightly greater than the modest colocalization present in undamaged cells (please see our response to point REV2-5). These new data address the reviewer's initial concern that the different Pol IV localization patterns for NFZ and MMS might be an artifact due to the different exposure times used initially.

REV2-2

This relates to the new localisation pattern for pol IV that appears after NFZ treatment. The authors discuss this further, but the discussion adds little to the picture. The localisation pattern they observe does not depend on interaction with beta-clamps or the catalytic activity of pol IV. The simplest interpretation is that observed localisation pattern doesn't tell us anything about the function of pol IV. The authors certainly don't provide any data to link this localisation pattern with biological function, such as NFZ survival assays. What does recruitment mean if it is not a step that precedes activity? What is presented in this manuscript could equally be described as "sticking to stuff". Yes, the localisations of RecA-GFP and DnaQ-PAmCherry show some similarities to the pol IV-PAmCherry pattern, but this may also be coincidence. Without data that link function to localisation, no meaning can be extracted from the localisation pattern.

We have done a number of experiments to try to understand this localization pattern, and a number of control experiments (including new experiments and analyses prompted by the reviewer) to demonstrate that it is Pol IV-dependent and DNA damage-dependent—in other words, to show that it is experimentally robust and specific to treatment with model damaging agents long used to study TLS generally and Pol IV specifically. Ultimately we were unable to identify the molecular interactions responsible for this localization, although we did show that it is independent of β -clamp binding and that the Pol IV N-terminus is sufficient. We note that we do observe β -dependent Pol IV recruitment in MMS-treated cells, which serves as further confirmation that our Pol IV-PAmCherry fusion is functional and that the localization in NFZ-treated cells is not simply an artifact of the fluorescent protein.

Although Pol IV has been implicated in other DNA damage tolerance and repair pathways, which we discuss in the manuscript, the biochemical characterization of these processes is largely unexplored. Thus it is not a matter of simply making a known mutation to determine whether the Pol IV localization is due to involvement in one of these pathways. In light of the amount of data already contained in this manuscript, however, we believe that further work on this localization pattern, while important, is beyond the scope of this study. Further, we disagree that the behavior of a significant fraction of the Pol IV molecules in the cell is irrelevant. If we have demonstrated adequately that this response is not an experimental artifact, then we feel it is worthy of publication.

REV2-3

Reviewer #2 raised concerns about cell size and viability upon treatment with MMS and NFZ. The changes made to the manuscript adequately address these concerns. The cells are too sick to filament, but this clearly does not affect their viability once removed from MMS/NFZ.

We are glad that our new experiments and analyses have addressed this concern.

REV2-4

This relates to the lifetimes of pol IV foci. The authors' interpretation is that most MMS-induced foci are due to non-TLS events or events that do not involve extensive synthesis (presumably TLS events with short processivity). No attempt was made to test either of these ideas using mutants impaired for beta-binding or catalytic activity. Again, without experimental links to biological function, the authors are speculating. The NFZ-induced are long-lived. Does this require catalytic activity? Or binding to beta-clamps? The authors speculate that the long-lived binding might reflect involvement in pathways such as recombination and transcription-coupled TLS. Once again the authors' data do not actually teach us anything about the biological function of pol IV.

We have added a new supplemental figure, Supplementary Fig. 15, showing the binding lifetime curves for WT and mutant Pol IV proteins in undamaged cells, cells treated with 100 μ M NFZ for 1 h, and cells treated with 100 mM MMS for 20 min. In addition, we have carried out a new experiment in which we characterize the localization and binding lifetime of the catalytically inactive Pol IV-D103N mutant in MMS-treated cells.

In MMS-treated cells, the binding lifetime of the clamp-binding deficient Pol IV^{R,C} mutant is statistically indistinguishable from that of the WT protein. While that might seem surprising, we note that there is very little difference in the Pol IV^{WT} binding lifetime in undamaged and MMS-treated cells; in other words, the damage-induced, β -dependent binding events in MMS-treated cells occur on a similar timescale to the non-specific background Pol IV binding events. Even

though abrogation of clamp-binding in the Pol IV^{R,C} strain eliminates the β -dependent localizations in response to damage, the background of non-specific Pol IV binding with similar lifetime remains.

Prompted by the reviewer's comment, however, we have now examined the binding lifetime of the catalytically inactive Pol IV-D103N mutant in MMS-treated cells. Interestingly, we find a modest (~ 13%) but statistically significant increase in lifetime in comparison to Pol IV^{WT}. It has previously been proposed that Pol IV-D103N becomes "locked" at a DNA lesion, unable to complete the catalytic cycle¹. Therefore, if most of the β -dependent binding events in MMS-treated cells represented active synthesis, we might expect a larger increase in the Pol IV-D103N binding lifetime. This result supports our hypothesis that the majority of Pol IV localizations in MMS-treated cells represent transient binding to clamps and not active synthesis. Additionally, we find that the number of binding events for the Pol IV-D103N mutant is the same as for Pol IV^{WT} in MMS-treated cells (Supplementary Table 6), and that Pol IV-D103N colocalizes with the replication fork in MMS-treated cells, albeit less strongly than does Pol IV^{WT} (Fig. 5d). These results confirm that catalytic activity is not required for Pol IV recruitment upon MMS treatment.

In the case of NFZ treatment, we find very little difference in the binding lifetime among the different mutants, with the exception of a statistically significant increase of 19% in the lifetime of Pol IV^{CD} mutant relative to Pol IV^{WT}. Since the Pol IV localization patterns are similar for all the mutants examined, it may not be surprising that the binding lifetimes are also similar. Although we are not able to definitively assign the long-lived Pol IV binding in NFZ-treated cells to other pathways, we believe that such work goes well beyond the scope of this paper, especially since the role of Pol IV in these other pathways is still poorly described.

REV2-5

This point was originally about the analysis of colocalisation. The authors now bring in the issue of the number of pol IV molecules within enrichment sites. In doing so they raise an even more important point relating to over-interpretations of data.

I'll deal with the colocalisation distances first. Yes the authors explain that the dip at zero arises from several factors, but what remains missing is what the shapes of the plots in Figure 3 actually mean. What would a plot of colocalisation distances look like for something that is tightly colocalised? And by that I mean something MEASURED under similar circumstances to the pol IV-SSB data, i.e. same pair of fluorophores, same imaging conditions etc. What about a plot for two things that are not colocalised? What about a pair of proteins that both localise to nucleoid, but do not otherwise interact with each other? Without these kinds of calibration points, it is impossible to interpret the pol IV-SSB data.

To further strengthen our conclusions and aid interpretation, we have added several new experiments (described below) as well as a more quantitative colocalization analysis. This radial distribution function analysis normalizes the measured distribution of Pol IV-SSB distances by a simulated random distribution that is generated based on the number of localizations in each cell and the specific cell geometry^{2,3}. In the resulting radial distribution curve, $g(r)$, a value of 1 represents no enrichment relative to a random localization distribution at a separation distance r , with increasing values corresponding to positive enrichment. This approach accounts for the geometry of observed (and not simulated) cells, and importantly corrects for differences in cell size between different strains or treatment conditions.

We have used this approach to reproduce colocalization data for *E. coli* ParC-PAmCherry and MukB-mYPet, which are modestly colocalized, as reported previously by the laboratories of Achilles Kapanidis and David Sherratt.² After obtaining the strain from the authors, we found excellent agreement with their published ParC-MukB $g(r)$ curve, including the maximum magnitude of enrichment, the radial distance at which the $g(r)$ curve crosses zero, and the percentage of ParC localizations within 200 nm of a MukB focus:

Figure from Zawadzki, P. *et al. Cell Rep.* 13, 2587 (2015) (Fig. 2B).

Figure from Thrall *et al.*, current submission (Supplementary Fig. 3b)

The ParC-MukB data serve as a positive control for weak colocalization. We were not able to find a literature report of strong colocalization using the same PAmCherry-mYPet fluorescent protein pair, but we note that the Pol IV-SSB colocalization that we see in MMS-treated cells, with a maximum $g(r)$ value of approximately 10, is more comparable to the strong colocalization between LacI-PAmCherry and Mall-GFP measured in a recent paper (compare Fig. 5C in Garza de Leon, *et al.* to Fig. 3c in our manuscript)³.

As requested by the reviewer, we have also added new data (Supplementary Fig. 3d) to show the apparent replisome colocalization for a nucleoid-associated protein, a previously-characterized HU-PAmCherry fusion⁴. The localization of HU-PAmCherry averaged over many cells is very strongly peaked at the quarter-cell position in undamaged cells, closely matching the average localization of SSB-mYPet. Nonetheless, we do not see significant intra-cell colocalization between HU and SSB at short distances ($g(r) < 150$ nm), as expected. The $g(r)$ value approaches 1 as r approaches zero. The $g(r)$ curve increases at intermediate separations, which reflects the fact that both HU and SSB localize in similar cellular regions. This result confirms that strong nucleoid localization alone does not result in significant colocalization at short distances.

Finally we have included colocalization analysis for the ϵ -PAmCherry SSB-mYPet strain in undamaged cells (Supplementary Fig. 3d). Here, we see stronger colocalization than for ParC-MukB, but weaker than for Pol IV-SSB in MMS-treated cells. This result demonstrates that we can detect the colocalization of 3 – 6 copies of ϵ in replication foci relative to the non-specific binding of the several hundred copies present in the cell⁵.

This new radial distribution analysis has revealed several features that were not apparent in our original colocalization analysis. First, we observe modest but distinct Pol IV-SSB colocalization in undamaged cells, which is dependent on β -clamp binding (compare Pol IV^{WT} and Pol IV^{R,C} in Fig. 3b). This colocalization is lost in cells treated with NFZ for 1 h (Fig. 3c). At the 20 min NFZ time point, the colocalization is still present, and may be very modestly enhanced relative to undamaged cells (Fig. 3d); again this colocalization is dependent on clamp binding (Fig. 5c). This analysis strengthens our conclusion that Pol IV does not colocalize strongly with replication forks in undamaged or NFZ-treated cells, and it allows us to quantify the increase in colocalization observed upon MMS treatment. We believe that these new experiments and analyses significantly strengthen our manuscript and we thank the reviewer for prompting us to improve this aspect of the paper.

Now the number of pol IV molecules at enrichment sites. The authors' claims are predicated on terrible use of extrapolation. The authors measure the number of static foci in their region-of-interest (fine), and then compare this to the number of static foci in fixed cells (not fine). Based on experience I think fixing is likely to severely reduce the number of fluorophores that are activateable within the cells. The authors have provided no evidence to the contrary. Next the authors' compare what they claim to be the percentage of molecules that form static foci in the ROI to the literature value of 2000 pol IV molecules per cell – really, really not OK! Firstly, this number was derived from Western blot, which are far from precise and have in many cases been shown to be inaccurate. Secondly, the authors have not compared the number of pol IV-PAmCherry molecules in the imaging strain with the number of pol IV molecules in wild type cells. Put together, these issues could impact the authors' estimates for the number of molecules in the ROI by two or three orders of magnitude! This estimate is an even worse basis for their

claim that pol IV acts at gaps behind the fork than the colocalisation measurements! In their rebuttal the authors claim that the Foster group's study used a less sensitive measurement than theirs. They certainly would have noticed foci containing 150 dinB-eYFP molecules!

Although this calculation was intended as a rough estimate we think it is unlikely to be off by the two to three orders of magnitude suggested by the reviewer. In response to the reviewer's concerns, we have quantified the two factors identified by the reviewer, which has led us to lower our enrichment estimate by one order of magnitude.

First, we quantified the effect of fixation. From PALM experiments using short integration times and matched imaging conditions, we measure a roughly 40% reduction in the number of tracks in fixed cells in comparison to undamaged cells. This is broadly consistent with a reduction of 20% reported for a different PAmCherry fusion in fixed cells⁶.

To estimate the expression level of the Pol IV-PAmCherry fusion in our imaging strain without relying on Western blot data, we measured the copy number of a well-characterized Pol I-PAmCherry fusion in fixed cells and compared it to the Pol IV copy number measured under matched conditions. (We expect that fixation should affect the two measurements comparably because the fluorescent protein is the same.) Based on the absolute number of Pol I copies measured by imaging⁶, we estimate an actual expression level of approximately 250 – 300 Pol IV-PAmCherry molecules in our imaging strain (see Supplementary Note 7).

Adjusting our calculation to account for these two effects results in a reduction of one order of magnitude in the estimated Pol IV copy number in NFZ- and MMS-treated cells, to approximately 11 copies in NFZ-treated cells and 6 in MMS-treated cells. We have modified Supplementary Note 7 to emphasize that this copy number is just an estimate and to describe more clearly the uncertainties in the value. In light of the reduction in the estimated enrichment and the reviewer's concerns, we have modified the text in the discussion section to temper the claim and describe the competing interpretation:

We can make a rough estimate of the number of Pol IV molecules that localize in MMS-treated cells based on the percentage of static Pol IV molecules and the copy number of the Pol IV-PAmCherry fusion (see Supplementary Note 7), and we conservatively estimate that there are 6 copies in the region of enrichment. Given the magnitude of this enrichment, it is possible that the observed Pol IV molecules are exclusively binding the two replisome-associated copies of β , one on the leading and one on the lagging strand, at each replication fork. Since this would require high Pol IV occupancy on the replisome-associated clamps, however, we favor the alternative explanation that Pol IV is also interacting with β clamps behind the fork.

Finally, we apologize that our statement about the Foster lab study was unclear. We were only responding to the reviewer's point (#3 in the original review) that the Foster lab did not observe Pol IV foci in undamaged cells, which was being contrasted to our observation of static binding events even in the absence of DNA damage, leading to the reviewer's concern that many of the cells being imaged were not viable. We were simply pointing out that PALM imaging is more sensitive than conventional fluorescence imaging and better able to resolve relatively short binding events of single molecules. Certainly we agree that the Foster lab would have detected long-lived foci containing many copies of Pol IV but, as our measurements confirm, such foci do not exist in undamaged cells.

References cited:

1. Heltzel, J. M. H., Maul, R. W., Scouten Ponticelli, S. K. & Sutton, M. D. A model for DNA polymerase switching involving a single cleft and the rim of the sliding clamp. *Proc. Natl. Acad. Sci. U. S. A.* **106**, 12664–12669 (2009).
2. Zawadzki, P. *et al.* The localization and action of Topoisomerase IV in Escherichia coli chromosome segregation is coordinated by the SMC complex, MukBEF. *Cell Rep.* **13**, 2587–96 (2015).
3. Garza de Leon, F., Sellars, L., Stracy, M., Busby, S. J. W. & Kapanidis, A. N. Tracking low-copy transcription factors in living bacteria: the case of the lac repressor. *Biophys. J.* **112**, 1316–1327 (2017).
4. Wang, S., Moffitt, J. R., Dempsey, G. T., Xie, X. S. & Zhuang, X. Characterization and development of photoactivatable fluorescent proteins for single-molecule-based superresolution imaging. *Proc. Natl. Acad. Sci. U. S. A.* **111**, 8452–7 (2014).
5. Reyes-Lamothe, R., Sherratt, D. J. & Leake, M. C. Stoichiometry and architecture of active DNA replication machinery in Escherichia coli. *Science* **328**, 498–501 (2010).
6. Uphoff, S., Reyes-Lamothe, R., Garza de Leon, F., Sherratt, D. J. & Kapanidis, A. N. Single-molecule DNA repair in live bacteria. *Proc. Natl. Acad. Sci. U. S. A.* **110**, 8063–8 (2013).

Reviewers' Comments:

Reviewer #2:

Remarks to the Author:

This third version of the Thrall et al. manuscript is much improved again, but still some serious issues remain. The colocalisation analysis is done in a much more meaningful way. Catalytically dead and clamp binding-defective mutants of Pol IV show few differences from wild-type Pol IV with respect to cellular localisation, binding lifetimes and colocalisation with SSB, which is unfortunate. I leave it to the editor to decide whether observations that are carefully documented, but poorly correlate with function, are suitable for publication in Nature Communications.

Two other major issues remain. The first relates to the rough estimate of the number of molecules of Pol IV appearing near replisomes (actually near SSB foci, see below). As the authors themselves point out, their estimates are rough –really rough. There are so many steps involved (each carrying their own errors and assumptions) that I simply cannot give any weight to these numbers. While it would be very interesting to know how many Pol IV molecules are bound at replisomes, these rough estimates do more harm than good. This is a matter of scientific validity, not impact. The authors should find a more direct way to measure these numbers or remove them from the manuscript entirely.

The second major issue relates to the colocalisation. The authors now present some beautiful data, which show clear changes upon MMS treatment (I'm going to ignore the NFZ data as nothing seems to effect Pol IV behaviour under those conditions). The authors replicate an example of 'weak' colocalisation – good. They also show what happens when you measure distances between SSB and the more generic DNA binder HU – also good. Colocalisation between SSB and epsilon looks weak – not sure how this can be. Colocalisation between SSB and Pol IV looks weak in the absence of damage, but very strong in the presence of MMS! Much stronger than epsilon-SSB! The authors nevertheless continue to refer to the colocalisation as weak and cite this as evidence for Pol IV acting in gaps behind forks. Their data say one thing, but the authors themselves say another.

The fact that the colocalisation between epsilon and SSB is weak suggests that SSB may be a poor marker for replisome position under these conditions. This may indeed be the case if MMS treatment leads to gaps being created. SSB is apparently a great marker for Pol IV binding sites created in response to MMS exposure, although it is impossible to tell exactly how strong the colocalisation really is without a positive control.

Reviewer #3:

Remarks to the Author:

Focusing on the concerns raised by reviewer #2, I conclude that they do not represent a significant problem to reach the main conclusions drawn by the authors. My reasoning is as follows:

1. The reviewer first raises the concern that the effect of different mutations on Pol IV behaviour is minimal. I agree with the authors in that even though some of the changes are subtle, they are able to show significant difference between them.
2. The reviewer points to two major concerns, the first of which is on the estimation of copy numbers for bound Pol IV. Here the authors could have done a better job at describing what they did, as the procedure followed is not standard in the field. For example, in their description they mention: "We calculated the mean number of static Pol IV molecules in the region of interest by fitting the spike in the subtracted localization distribution to a Gaussian. This procedure gave a mean of 2.6 molecules per cell". However, it is not clear to me how do they obtain the number of 2.6 molecules. Did they base this on their estimation of the total number of Pol IV in the cell for the untreated cells? However, the copy number per cell was later used to normalize their estimate,

so if they used it in the first step, it would then represent a circular argument. They should clarify this.

In addition, they should illustrate the procedure in a figure and they should estimate the standard error for those numbers.

I agree that this is a minor point and that the authors might decide to remove their statements on copy number without affecting the main conclusions of the paper.

3. The reviewer's second major concern is on the colocalization of Pol IV with the replisome. For this analysis the authors followed procedures used by other groups. As the authors mention, it is difficult to compare the results for different proteins as they will depend on multiple variables. For example, I think that the relatively low enrichment for the epsilon subunit of the Pol III with SSB spots can be partly explained by transient binding of epsilon at sites outside of the replication fork. Different proteins will have different tendencies for transient binding. The authors could have considered using only tracks with multiple localizations, which should result in a greater number of fork-associated copies of epsilon, but this choice of parameters during the analysis may not work for Pol IV.

However, the differences for the localization of Pol IV relative to SSB across conditions are clear, which is the most important point. I do not think they require to characterize a positive control with a different protein to demonstrate that such differences exist.

We thank the reviewers for their careful consideration of our manuscript. We address the detailed reviewer comments below. Our responses are in blue.

We have highlighted changes made in response to reviewer comments in the main text, although the only changes made were related to the removal of the Pol IV copy number estimate. We did not highlight changes in the Supplementary Information, as the only change was the removal of Supplementary Note 7, which had contained the details of the copy number estimate.

Reviewers' comments:

Reviewer #2 (Remarks to the Author):

This third version of the Thrall et al. manuscript is much improved again, but still some serious issues remain. The colocalisation analysis in is done in a much more meaningful way. Catalytically dead and clamp binding-defective mutants of Pol IV show few differences from wild-type Pol IV with respect to cellular localisation, binding lifetimes and colocalisation with SSB, which is unfortunate. I leave it to the editor to decide whether observations that are carefully documented, but poorly correlate with function, are suitable for publication in Nature Communications.

We do not believe that the reviewer's overall characterization of our results is fully accurate. The reviewer states that we see few differences for the Pol IV mutants relative to the WT protein. In the case of MMS, however, we see statistically significant and in some cases quite large changes in the number and lifetime of Pol IV binding events (Supplementary Figs. 4d and 15c and Supplementary Table 6), the cellular localization (Fig. 5b), and the Pol IV-SSB colocalization (Fig. 5d) for the catalytically inactive and clamp-binding deficient mutants. For NFZ-treated cells, we do see fewer effects for these Pol IV mutants, although notably we do observe a loss of Pol IV-SSB colocalization for the clamp-binding deficient mutant in the early stage of the damage response (Fig. 5c). We also observe a loss of Pol IV-SSB colocalization for the clamp-binding deficient mutant in undamaged cells (Fig. 3b). Why there are differences in the response to NFZ and MMS is an important question for future studies, but regardless we do observe clear effects for the catalytically inactive and clamp-binding deficient Pol IV mutants under many conditions.

Two other major issues remain. The first relates to the rough estimate of the number of molecules of Pol IV appearing near replisomes (actually near SSB foci, see below). As the authors themselves point out, their estimates are rough –really rough. There are so many steps involved (each carrying their own errors and assumptions) that I simply cannot give any weight to these numbers. While it would be very interesting to know how many Pol IV molecules are bound at replisomes, these rough estimates do more harm than good. This is a matter of scientific

validity, not impact. The authors should find a more direct way to measure these numbers or remove them from the manuscript entirely.

In the previous round of review, we performed several careful experiments to address points raised by the reviewer and reduce the uncertainty in the Pol IV copy number estimate. Although Reviewer #2 is still concerned about the estimate, both reviewers are in agreement that this copy number estimate is a minor point that does not affect the main conclusions of the manuscript, and we share that view. For that reason, and to avoid unnecessary confusion, we have decided to remove the estimate from the manuscript entirely.

The second major issue relates to the colocalisation. The authors now present some beautiful data, which show clear changes upon MMS treatment (I'm going to ignore the NFZ data as nothing seems to effect Pol IV behaviour under those conditions). The authors replicate an example of 'weak' colocalisation – good. They also show what happens when you measure distances between SSB and the more generic DNA binder HU – also good. Colocalisation between SSB and epsilon looks weak – not sure how this can be. Colocalisation between SSB and Pol IV looks weak in the absence of damage, but very strong in the presence of MMS! Much stronger than epsilon-SSB! The authors nevertheless continue to refer to the colocalisation as weak and cite this as evidence for Pol IV acting in gaps behind forks. Their data say one thing, but the authors themselves say another.

Regarding the colocalization analysis, the reviewer appears to be confusing our discussion of the NFZ and MMS results. We never refer to the Pol IV-SSB colocalization in MMS-treated cells as “weak,” nor do we offer the degree of colocalization as support for the idea that Pol IV is acting at gaps behind the fork. The colocalization in undamaged and NFZ-treated cells is weak, and we describe it as such, but we agree with the reviewer that the Pol IV-SSB colocalization is strong in MMS-treated cells. (See for example page 15 in the discussion, where we state: “In MMS-treated cells, we observed strong Pol IV recruitment to replication forks marked by SSB foci...”)

The fact that the colocalisation between epsilon and SSB is weak suggests that SSB may be a poor marker for replisome position under these conditions. This may indeed be the case if MMS treatment leads to gaps being created. SSB is apparently a great marker for Pol IV binding sites created in response to MMS exposure, although it is impossible to tell exactly how strong the colocalisation really is without a positive control.

Although the reviewer appears to expect ϵ -SSB colocalization even stronger than the level that we observe, it's important to remember that the apparent colocalization includes several effects, including the copy numbers of bound vs. unbound molecules and the rate of non-specific binding elsewhere in the cell. For this reason, comparison of the $g(r)$ curves for different proteins is

challenging. Also, we note that the ϵ -SSB measurements were performed in undamaged cells, where there should not be large ssDNA gaps away from the replication fork.

More generally, SSB has been used in numerous studies as a replisome marker (for example, Reyes-Lamothe, R. *et al.*, *Cell*, 2008 and Mangiameli, M. *et al.*, *PLoS Genet.*, 2017) , and we do not believe that it is a poor proxy for replisome position. We already accounted for this possibility, however, by comparing the number of SSB-mYPet and ϵ -mYPet foci in MMS-treated cells (Supplementary Table 4), which are in excellent agreement. Likewise, the average cellular localization for SSB-mYPet and ϵ -mYPet in MMS-treated are almost identical. These results indicate that SSB is a reliable marker for the replisome, even in MMS-treated cells, and that SSB foci are marking active replication forks and not ssDNA gaps.

Reviewer #3 (Remarks to the Author):

Focusing on the concerns raised by reviewer #2, I conclude that they do not represent a significant problem to reach the main conclusions drawn by the authors. My reasoning is as follows:

1. The reviewer first raises the concern that the effect of different mutations on Pol IV behaviour is minimal. I agree with the authors in that even though some of the changes are subtle, they are able to show significant difference between them.

We are glad that the reviewer agrees with our assessment of these results.

2. The reviewer points to two major concerns, the first of which is on the estimation of copy numbers for bound Pol IV. Here the authors could have done a better job at describing what they did, as the procedure followed is not standard in the field. For example, in their description they mention: “We calculated the mean number of static Pol IV molecules in the region of interest by fitting the spike in the subtracted localization distribution to a Gaussian. This procedure gave a mean of 2.6 molecules per cell”. However, it is not clear to me how do they obtain the number of 2.6 molecules. Did they base this on their estimation of the total number of Pol IV in the cell for the untreated cells? However, the copy number per cell was later used to normalize their estimate, so if they used it in the first step, it would then represent a circular argument. They should clarify this.

In addition, they should illustrate the procedure in a figure and they should estimate the standard error for those numbers.

I agree that this is a minor point and that the authors might decide to remove their statements on copy number without affecting the main conclusions of the paper.

Please see our response to the comments of Reviewer #2 above.

3. The reviewer's second major concern is on the colocalization of Pol IV with the replisome. For this analysis the authors followed procedures used by other groups. As the authors mention, it is difficult to compare the results for different proteins as they will depend on multiple variables. For example, I think that the relatively low enrichment for the epsilon subunit of the Pol III with SSB spots can be partly explained by transient binding of epsilon at sites outside of the replication fork. Different proteins will have different tendencies for transient binding. The authors could have considered using only tracks with multiple localizations, which should result in a greater number of fork-associated copies of epsilon, but this choice of parameters during the analysis may not work for Pol IV.

However, the differences for the localization of Pol IV relative to SSB across conditions are clear, which is the most important point. I do not think they require to characterize a positive control with a different protein to demonstrate that such differences exist.

We agree with the reviewer's interpretation of the ϵ -SSB colocalization results, and share the reviewer's opinion that the changes in Pol IV-SSB colocalization across different conditions are clear without further analysis of the ϵ -SSB positive control.